# Mapping of in vivo cleavage sites uncovers a major role for yeast RNase III in regulating protein-coding genes

Lee-Ann Notice-Sarpaning[1,2], Mathieu Catala[3], Catherine Stuart[1], Sherif Abou Elela[3], Ambro van Hoof[1,2]*†

[1]Department of Microbiology and Molecular Genetics and University of Texas Health Science Center at Houston, Houston, United States; [2]UT MD Anderson Cancer Center UTHealth Houston Graduate School of Biomedical Sciences, University of Texas Health Science Center at Houston, Houston, United States; [3]Département de Microbiologie et d'Infectiologie, Université de Sherbrooke, Sherbrooke, Canada

**\*For correspondence:**
ambro.van.hoof@uth.tmc.edu

**Present address:** †Department of Molecular Genetics, Ohio State University, Columbus, United States

**Competing interest:** The authors declare that no competing interests exist.

## eLife Assessment

This **valuable** study expands the inventory of polyadenylated RNAs cleaved by the double-stranded RNA endonuclease Rnt1 in budding yeast, using **solid** methodology based on high-throughput sequencing. Previous studies had anecdotally discovered mRNA substrates, and this global characterization is comprehensive with multiple complementary controls. This study sets the stage for deeper investigations into the biological function of Rnt1 and substrate cleavage.

**Abstract** A large fraction of newly transcribed RNA is degraded in the nucleus, but nuclear mRNA degradation pathways remain largely understudied. The yeast nuclear endoribonuclease Rnt1 has a well-characterized role in the maturation of many ncRNA precursors. However, the scope and consequence of its function in mRNA degradation pathways are much less defined. Here, we take a whole-transcriptome approach to identify Rnt1 cleavage sites throughout the yeast transcriptome in vivo, at single-nucleotide resolution. We discover previously unknown Rnt1 cleavage sites in many protein-coding regions and find that the sequences and structures necessary for cleavage mirror those required for the cleavage of known targets. We show that the nuclear localization of Rnt1 functions as an additional layer of target selection control, and that cleaved mRNAs are likely exported to the cytoplasm to be degraded by Xrn1. Further, we find that several cleavage products are much more abundant in our degradome sequencing libraries than decapping products, and strikingly, mutations in one Rnt1 target, *YDR514C*, suppress the growth defect of an *RNT1* deletion. Overexpression of *YDR514C* results in slow growth, further suggesting that Rnt1 may limit the expression of *YDR514C* to maintain proper cell growth. This study uncovers a broader target range and function for the well-known RNase III enzyme.

## Introduction

RNA degradation is an essential process for maintaining cellular homeostasis, regulating both the quantity and quality of RNAs in the eukaryotic cell. RNA degradation typically occurs by exoribonuclease attack from either end of the transcript, but in some cases, is initiated by endoribonuclease cleavage at internal sites. Endoribonuclease cleavage of certain RNAs modulates important cellular processes, including differentiation and stress response. As such, mutations in endoribonucleases have been implicated in many human diseases. However, eukaryotic endoribonucleases remain largely

uncharacterized, and in most cases of disease, specific RNA transcripts that fail to be cleaved have not been identified, highlighting the need for systematic identification of endoribonuclease targets.

Indeed, recent transcriptome-wide studies have uncovered non-classical RNA targets for well-studied endoribonucleases. One notable example is the human RNase III enzyme, Drosha. Until recently, Drosha's only known function has been the maturation of micro-(mi)RNAs (*Lee et al., 2003*). It is now clear, however, that Drosha directly cleaves other classes of RNAs, including mRNAs (*Han et al., 2009*; *Karginov et al., 2010*; *Knuckles et al., 2012*), long non-coding (lnc)RNAs (*Dhir et al., 2015*), and exogenous viral RNAs (*Lin and Sullivan, 2011*; *Shapiro et al., 2012*; *Shapiro et al., 2014*).

Like its distant metazoan homolog Drosha, the *Saccharomyces cerevisiae* RNase III enzyme, Rnt1, has also been shown to process ncRNAs, including small nuclear (sn)RNA and small nucleolar (sno) RNA precursors. snRNAs are essential components of the spliceosome, and snoRNAs guide the chemical modification of rRNA and snRNAs. The most abundant target of Rnt1, however, is rRNA. Although Rnt1 is not essential, a *rnt1Δ* strain is extremely slow-growing (*Chanfreau et al., 1997*; *Chanfreau et al., 1998b*), and this is presumed to be due to the requirement of Rnt1 in rRNA processing.

Rnt1 carries out 3′ end maturation of pre-rRNA by cleaving at a specific site downstream of the mature 25 S 3′ end in the 3′ external transcribed spacer (ETS; *Elela et al., 1996*). The U3 pre-snoRNA and the U1, U2, U4, and U5 pre-snRNAs are also cleaved to remove a 3′ extension downstream of their mature ends. At both pre-rRNA and pre-snRNA genes, co-transcriptional 3′ cleavage generates an entry site for the nuclear 5′–3′ exoribonuclease Rat1. Rat1-mediated digestion of the nascent RNA then triggers transcription termination (*Allmang et al., 1999*; *Allmang and Tollervey, 1998*; *El Hage et al., 2008*). Many pre-snoRNAs, on the other hand, are 5′ end processed (*Chanfreau et al., 1998b*). While some snoRNAs contain a trimethylated G (TMG) cap and do not require 5′ processing (*Chanfreau et al., 1998a*), others rely on Rnt1 to either remove a m⁷G-capped extension or to both remove the 5′ extension and liberate them from polycistronic gene arrays (*Chanfreau et al., 1998a*; *Chanfreau et al., 1998b*; *Qu et al., 1999*). Failure to remove pre-snoRNA m⁷G caps results in snoRNA mislocalization to the cytoplasm and also affects the processing of their 3′ ends (*Grzechnik et al., 2018*). Additionally, some intron-encoded pre-snoRNAs require Rnt1 cleavage for release from their host introns (*Ghazal et al., 2005*; *Giorgi et al., 2001*). While cleavage of pre-snRNAs and pre-snoRNAs is followed by Rat1 exoribonucleolytic digestion, the consequences differ; Rat1 processing produces mature 5′ ends of snoRNAs, but causes transcription termination of pre-snRNAs.

Rnt1 recognizes double-stranded (ds)RNA stem loops with a terminal single-stranded tetranucleotide loop containing the consensus sequence AGNN (*Catala et al., 2004*; *Elela et al., 1996*; *Nagel and Ares, 2000*). This tetraloop is the major determinant of Rnt1 substrate recognition as it dictates the three-dimensional conformation of the substrate and facilitates docking of Rnt1 (*Chanfreau et al., 2000*; *Lebars et al., 2001*; *Wu et al., 2001*). Specifically, the conserved A and G nucleotides in the first two positions of the tetraloop allow Rnt1 interaction with the loop (*Lebars et al., 2001*; *Wu et al., 2001*). Rnt1 also binds UGNN tetraloops, but with lower affinity, and can cleave 3- and 5-nt terminal loops with diverse sequences in vitro (*Gagnon et al., 2015*; *Lebars et al., 2001*). Rnt1 cleaves 14 nts upstream and/or 16 nts downstream of the tetraloop using its RNase III nuclease domain (*Chanfreau et al., 2000*). The sequences surrounding the tetraloop can impact the stability of the stem and affect Rnt1 binding, while sequences close to the cleavage site can affect the ability of Rnt1 to cleave (*Lamontagne et al., 2003*; *Lamontagne et al., 2004*). This results in Rnt1 binding and cleavage efficiency that is difficult to predict systematically (*Lamontagne et al., 2003*).

Cleavage results in 3′ fragments with a free 5′ monophosphate (*Court et al., 2013*; *Lamontagne et al., 2004*) that are substrates for Rat1 and/or the primarily cytoplasmic 5′–3′ exoribonuclease Xrn1 (*Kenna et al., 1993*; *Stevens, 1980*). Upon Rnt1 cleavage, Rat1 processes ncRNA precursor 5′ ends in the nucleus (*Chanfreau et al., 1997*; *Henry et al., 1994*), but when a Rnt1 site is artificially inserted into mRNAs, the resulting cleavage products are degraded from their 5′ ends predominantly by Xrn1 in the cytoplasm (*Meaux et al., 2011*). Rnt1-cleaved 3′ fragments, therefore, accumulate in a *rat1-ts xrn1Δ* strain background (*Geerlings et al., 2000*; *Petfalski et al., 1998*).

While the role of Rnt1 in RNA processing is well understood, much less is known about a potential function of this enzyme in RNA degradation. Nuclear RNA degradation by the exoribonucleases Rat1 (XRN2 in humans) and the RNA exosome has been extensively studied. These two RNases degrade many by-products of gene expression, including spliced out introns, the 5′ and 3′ ETS of rRNA, the RNA downstream of the cleavage and polyadenylation site, enhancer RNAs, promoter upstream

transcripts (PROMPTs), and cryptic unstable transcripts (CUTs). Another important class of substrates for these two RNases are misprocessed RNAs, which in some cases involves intricate recognition mechanisms. For example, recent work in human cells demonstrates that the combination of a 5′ splice site and poly(A) junction recruits the RNA exosome to incomplete mRNAs (*Soles et al., 2025*). Similarly, in yeast, aberrant mRNAs caused by defects in the THO complex are targeted to the RNA exosome (*Assenholt et al., 2008*). While the role of exoribonucleases in nuclear mRNA decay and quality control has been characterized, much less is known about the contributions of endoribonucleases. While the PIN-domain endoribonuclease Swt1 has been implicated in nuclear degradation of mRNAs that are not exported efficiently (*Skružný et al., 2009*), the repertoire of other potential nuclear endoribonucleases in these processes has not been fully investigated.

We therefore sought to characterize the repertoire of RNAs cleaved by Rnt1. Throughout the literature, a total of 102 Rnt1 cleavage sites have been implicated in ncRNA maturation, and many of these have been confirmed by multiple studies. In contrast, the 36 published Rnt1 cleavage sites in mRNAs have largely been identified in isolation and have not been confirmed by other studies. Studies have used in vitro cleavage to identify Rnt1 mRNA targets, but only a limited subset of these mRNAs have been shown to be cleaved in vivo. Additionally, some studies have reported changes in overall mRNA levels, but these include indirect effects. Genome-wide studies have identified a large number of mRNAs that are upregulated when Rnt1 is deleted but are not cleaved by the enzyme in vitro, while many other RNAs that are cleaved by the purified enzyme in vitro are not affected by the deletion of the enzyme in vivo. This raises questions about whether certain Rnt1 substrates might only be recognized in vivo. Here, we take an in vivo approach to identify bona fide Rnt1 cleavage sites throughout the yeast transcriptome, focusing our analysis on protein-coding regions. We find that Rnt1 directly cleaves several mRNAs with features resembling known targets and that cleaved mRNAs are subsequently degraded by Xrn1. We also show that Rnt1 regulation of one highly cleaved target is essential for maintaining proper cell growth.

## Results

### PARE identifies known Rnt1 cleavage sites and substrates

To identify in vivo Rnt1 cleavage sites throughout the yeast transcriptome, we used an RNA sequencing approach called parallel analysis of RNA ends (PARE). We have previously shown that PARE is effective for determining cleavage sites of tRNA splicing endonuclease (TSEN), while RNA-seq identifies hundreds of changes that do not include direct cleavage targets (*Hurtig et al., 2021*; *Hurtig and van Hoof, 2022*). Using PARE, we successfully identified mRNA targets of TSEN (*Hurtig et al., 2021*) and uncovered Dxo1 as a distributive exoribonuclease required for the final step in 25 S rRNA maturation (*Hurtig and van Hoof, 2022*).

To perform PARE, total RNA is isolated from cells, and an adapter is added onto free 5′ monophosphates using T4 RNA ligase (*Figure 1A*). The adapter is then used to build a sequencing library. Exposed 5′ phosphates result from decapping or endoribonuclease cleavage. Thus, RNAs can be sequenced from their 5′ monophosphate ends, and the precise positions of decapping or cleavage can be identified by accumulated reads starting at those specific nucleotide positions. Analysis of our previously published PARE data (*Hurtig et al., 2021*) revealed a peak of 5′ monophosphate ends at the known Rnt1 cleavage site in *BDF2*, suggesting that PARE could be used to identify additional Rnt1 sites. Further, comparison of PARE data using poly(A)$^+$ and poly(A)$^-$ RNA from strains lacking either the primarily cytoplasmic Xrn1 (*xrn1Δ*; *Hurtig et al., 2021*; *Hurtig and van Hoof, 2022*) or both Xrn1 and nuclear Rat1 (*rat1-ts xrn1Δ*; unpublished) revealed that known Rnt1 sites can be most prominently detected in the poly(A)$^+$ fraction of a *rat1-ts xrn1Δ* strain. PARE on the poly(A)$^-$ fraction predominantly detected the mature 5′ monophosphate ends of snoRNAs, and PARE on the poly(A)$^+$ fraction from an *xrn1Δ*-only strain weakly identified known Rnt1 sites. Thus, to identify Rnt1 cleavage sites, we used PARE to detect 5′ monophosphate ends that are present in a *RNT1 rat1-ts xrn1Δ* strain but absent from a *rnt1Δ rat1-ts xrn1Δ* strain (hereafter '*RNT1*' and '*rnt1Δ*', respectively, unless otherwise noted).

To distinguish bona fide Rnt1 cleavage events, we performed two replicates of PARE and focused on sites with ≥1 cpm (counts per million) in the *RNT1* strain in each replicate (*Figure 1B*; *Figure 1— figure supplement 1*). This cutoff eliminates very rare RNA 5″ ends. Additionally, we calculated a comPARE score, which is a log2 fold change modified to accommodate sites with 0 reads in the mutant

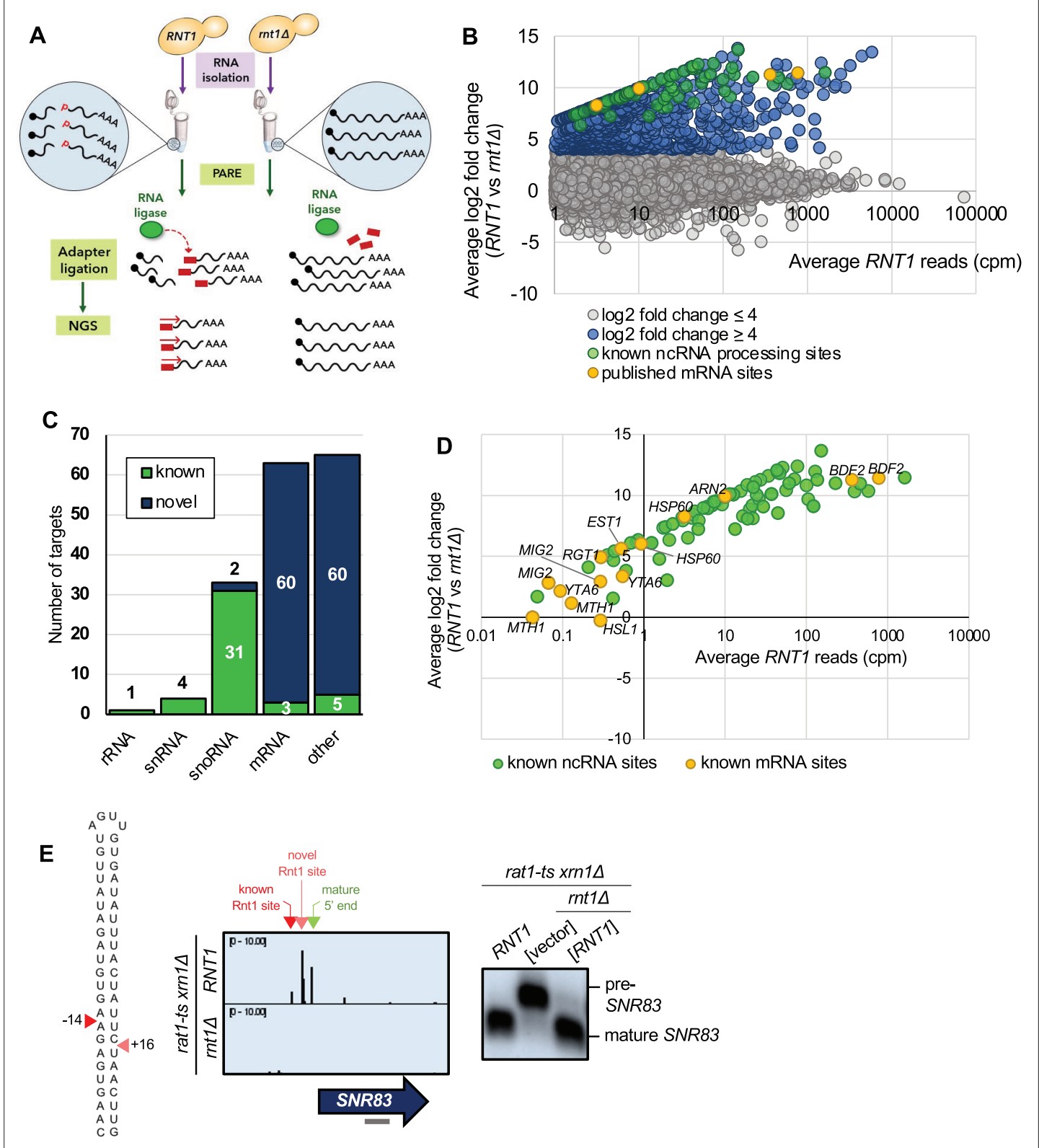

**Figure 1.** PARE identifies known Rnt1 cleavage sites and substrates. (**A**) Schematic of PARE workflow. Total RNA is isolated from *RNT1* and *rnt1Δ* strains. T4 RNA ligase ligates an adapter (red rectangle) onto exposed 5' phosphates (red Ps) resulting from cleavage or decapping. Next-generation sequencing is performed from the 5' adapter, resulting in reads that begin at the first nucleotide after the cleavage or decapping site. (**B**) >80,000 sites were detected with reads ≥1 cpm in the *RNT1* strain (x-axis). 496 of these were decreased in the *rnt1Δ* strain (log2(FC)>4 blue, green, and yellow

*Figure 1 continued on next page*

*Figure 1 continued*

dots). Known ncRNA processing sites are shown in green, and known mRNA sites are shown in yellow. Shown are the averages of two independent biological replicates. (**C**) 496 putative Rnt1 cleavage sites cluster into 166 different substrates. All known Rnt1 ncRNA targets are detected as well as two novel snoRNA targets. 63 mRNA targets are detected, of which 3 are known and 60 are novel. Other sites detected include intergenic, antisense, intronic, and 5′ and 3′ UTR sites. (**D**) PARE detects known Rnt1 cleavage sites in ncRNA targets, validating PARE as a reliable method for identifying novel Rnt1 cleavage sites. Some known mRNA sites are also detected, but most have reads <1 cpm in *RNT1*. (**E**) PARE precisely detects the known Rnt1 cleavage site in pre-*SNR83* (red arrowhead) located 61 nts upstream of its mature 5′ end (green arrowhead) on the 5′ side of an AGUU stem loop. PARE additionally reveals a novel site (pink arrowhead) located on the 3′ side of the stem. Structure of the stem loop and IGV PARE screenshot are shown. Northern blot using a probe that hybridizes to mature *SNR83* (grey bar) was performed in duplicate.

The online version of this article includes the following source data and figure supplement(s) for figure 1:

**Source data 1.** Original files for the northern blots in *Figure 1*.

**Source data 2.** Original files for the northern blots in *Figure 1* with labels of what parts were used in the figure.

**Figure supplement 1.** Bioinformatic pipeline for analysis of PARE data in Galaxy.

**Figure supplement 2.** Rnt1 cleavage site distribution.

**Figure supplement 3.** Examples of Rnt1 cleavage sites detected by PARE.

**Figure supplement 3—source data 1.** Original files for the northern blots in *Figure 1*.

**Figure supplement 3—source data 2.** Original files for the northern blots in *Figure 1* with labels of what parts were used in the figure.

(see Materials and methods and *Hurtig et al., 2021*). To identify Rnt1-dependent PARE peaks, we required a comPARE score of ≥4 (i.e. at least 16-fold difference in the number of reads) between *RNT1* and *rnt1Δ* in each replicate for a site to be considered a legitimate Rnt1 cleavage site (*Figure 1B*; *Figure 1—figure supplement 1*). The modified log2(FC) ≥4 was set empirically because it captured almost all known Rnt1 sites, and the vast majority of transcriptomic sites had a modified log2(FC) between –4 and 4. Very few sites had a modified log2(FC) <-4 (i.e. showed increased cleavage in the *rnt1Δ* strain), suggesting that this was an appropriate cutoff to eliminate noise. These two cutoffs identified 496 Rnt1 sites throughout the yeast transcriptome (*Supplementary file 1*). In many cases, two sites were separated by approximately 34 nts (*Figure 1—figure supplement 2*), as expected for cleavage 14 nts upstream and 16 nts downstream of a 4-nt loop. Other sites were separated by 1–5 nts (*Figure 1—figure supplement 2*), which can be explained by imprecise cleavage of Rnt1 or by slight degradation of the primary cleavage product by a 5″ exoribonuclease such as Dxo1 and/or Rat1 not fully inactivated by the temperature sensitive mutation. Overall, the 496 sites clustered into a total of 166 distinct RNA substrates (*Figure 1C*; *Supplementary file 1*).

Of the 166 putative substrates, 44 were known Rnt1 targets, most of which were ncRNAs, including pre-snRNAs, pre-snoRNAs, and the pre-rRNA 3′ ETS (*Figure 1C and D*; *Figure 1—figure supplement 3A*). This validates PARE and our cutoffs as a suitable approach for identifying Rnt1 sites and substrates. These sites could be visualized with Integrative Genomics Viewer (IGV), where the height of the peak at each position reflects the number of sequencing reads beginning at that position. For example, Rnt1 has recently been shown to cleave the snoRNA precursor pre-*SNR83* upstream of its mature 5′ end (*Grzechnik et al., 2018*). This site was detected in our PARE data (*Figure 1E*) where cleavage at that position can be observed in the *RNT1* strain, but not in *rnt1Δ*. Interestingly, we identified an additional Rnt1 site downstream of the known site. These two sites map on either side of an AGUU tetraloop, consistent with Rnt1's cleavage pattern. Northern blot using a probe that hybridized to the mature *SNR83* confirmed the Rnt1 cleavage detected by PARE. We observed processing to the mature snoRNA in the *RNT1* strain, as well as in a *rnt1Δ* strain containing *RNT1* on a plasmid, while the pre-snoRNA transcript accumulates in the *rnt1Δ* strain containing empty vector (*Figure 1E*). Thus, PARE confirmed pre-*SNR83* as a Rnt1 substrate, but also identified a second cleavage site.

Similarly, PARE identified previously unknown 5′ sites in the known pre-snoRNA substrates pre-*SNR39B*, pre-*SNR85*, pre-*SNR87*, and pre-*SNR81* (*Figure 1—figure supplement 3B*) and an unknown 3′ site in pre-*SNR17B* (*Figure 1—figure supplement 3C*). Additionally, although the snoRNA *SNR84* was thought to be capped (*Grzechnik et al., 2018*), we identified a peak of 5′ monophosphate ends at the mature 5′ end of *SNR84*, suggesting that the mature snoRNA is uncapped (*Figure 1— figure supplement 3D*). We detected additional sites –26 and –74 nts upstream of the mature 5′ end, and the sequence between these sites can fold into a stem loop structure with a bifurcated stem and an AGUA loop. All three sites disappeared in the *rnt1Δ* strain. These results suggest that an

uncapped snR84 snoRNA is made from a precursor with a transcription site at least 74 nts upstream. Rnt1 cleaves the 5′ extension of the pre-snoRNA at –26 and –74 nts, which is likely followed by Rat1 trimming to generate the mature snoRNA end. However, the processed uncapped *SNR84* we detect could co-exist with the previously reported capped *SNR84* snoRNA. Although we detected these novel ncRNA sites by PARE, most of the other ncRNA sites we detected precisely correspond to those in the published literature, demonstrating the reliability of PARE.

Additionally, PARE detected three previously published mRNA targets of Rnt1: *BDF2*, *HSP60*, and *ARN2* (*Figure 1C and D*). Although cleavage peaks were detected in an additional six known targets (*MIG2*, *EST1*, *YTA6*, *RGT1*, *MTH1*, and *HSL1*), these PARE peaks had low *RNT1* reads (<1 cpm) and/or did not meet our modified log2(FC) cutoff of ≥4 (*Figure 1D*). Thus, these six mRNAs may be targets of Rnt1 but are not the most prominent targets. *BDF2* is one of the most well-studied Rnt1 mRNA targets, and it was also one of the top targets in our dataset (second highest modified log2(FC); *Figure 2A*). PARE confirmed that *BDF2* was cleaved at the sites that have previously been described (*Figure 2A*). This mRNA adopts a secondary structure that facilitates the formation of a double-stranded stem loop within its ORF. The stem is capped by a UGAU tetraloop, instead of the more common AGNN, and Rnt1 sites are 14 nts away from the loop on the 5′ side and 16 nts away on the 3″ side (*Figure 2A*). In vivo cleavage was confirmed by northern blot, which showed the presence of the cleavage product (*Figure 2E*). *ARN2*, another Rnt1 mRNA target that has been previously described (*Lee et al., 2005*), was also shown to be cleaved in our PARE data (*Figure 1D*, *Figure 2—figure supplement 1*). Identification of these known mRNA targets further validates the use of PARE for determining novel Rnt1 cleavage sites in protein-coding regions.

## PARE identifies novel Rnt1 mRNA targets

In addition to the 44 known Rnt1 substrates, we also detected sites in 122 putative substrates. Although these included interesting novel targets such as pre-mRNA introns lacking snoRNAs (*Figure 1—figure supplement 3E*) and ncRNA transcripts derived from regions annotated as intergenic (*Figure 1—figure supplement 3F–H*), approximately half of the newly identified targets were mRNAs (i.e. protein-coding genes cleaved within their coding sequence [CDS]; *Figure 1C*). The CDS site with the third largest modified log2(FC) was in a novel substrate, *CAF4* (*Figure 2B*). *CAF4* encodes the Ccr4-associated factor 4 (Caf4), whose function is poorly understood. We found that *CAF4* mRNA was cleaved on both sides of a stem loop containing the canonical AGNN tetraloop sequence (AGGA), with an average 1000-fold difference (modified log2(FC)=10) in cleavage between the *RNT1* and *rnt1Δ* strains at the upstream site, and a 2500-fold difference at the downstream site (modified log2(FC)=11). The cleavage sites map to 15 nts upstream and 16 nts downstream of the tetraloop. Considering that the 15 nts upstream includes one bulged nucleotide, this resembles the typical Rnt1 cleavage pattern that results in a 2-nt overhang on the 3′ side of the stem loop (*Court et al., 2013*; *Lamontagne et al., 2004*). We also validated in vivo cleavage of this target by northern blot (*Figure 2E*).

Interestingly, other top targets *YDR514C* (highest modified log2(FC); *Figure 2C*) and *MTM1* (tenth highest log2(FC); *Figure 2D*) did not have canonical stem loops but instead were predicted to have more complex structures that included an AGNN loop. *YDR514C* had an AGGU tetraloop on a bulged stem (*Figure 2C*). Remarkably, there is still a 3′ 2-nt overhang along the double-stranded region of the *YDR514C* stem loop, consistent with cleavage by Rnt1. Unlike many Rnt1 mRNA substrates, we detected only a single site in *MTM1* (*Figure 2D*). *MTM1* has an AGGU stem loop with two other stem loops on either side, all radiating from a central bulge. While it is possible that *MTM1* may fold differently in vivo compared to the predicted RNA structure, one explanation for the single Rnt1 cleavage site could be that the two double-stranded regions on either side of the AGGU stem dictate that only one site can be cleaved. This single site still resembles the typical Rnt1 cleavage pattern by being 16 nts 3′ of the tetraloop. Thus, some Rnt1-cleaved mRNAs have variant stem structures that may, in part, be why they have not been previously identified.

Other highly cleaved, novel mRNA substrates with canonical AGNN loops included *TCB1*, *YER145C-A*, *PAN6*, *AVT1*, and *YPL277C* (*Figure 2—figure supplement 1*). Interestingly, although *YPL277C* and *YOR389W* are recent evolutionary duplicates, there was no PARE signal detected in *YOR389W* (*Figure 2—figure supplement 1C*). These mRNAs are 96% identical, with only a single-nucleotide difference in the 90-nucleotide stem loop. This one nucleotide difference disrupts a base

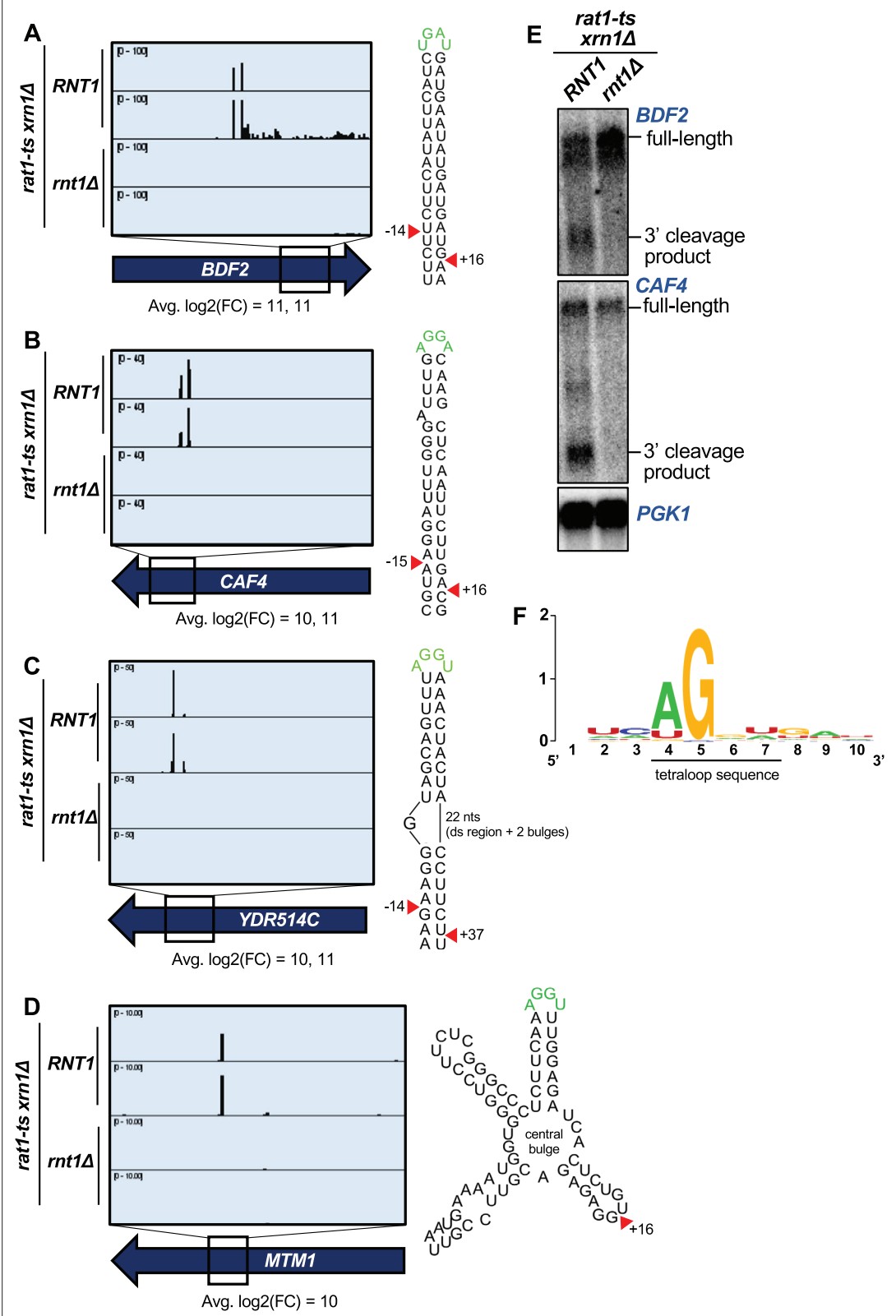

**Figure 2.** PARE identifies novel Rnt1 mRNA targets. (**A–D**) PARE screenshots of Rnt1-cleaved mRNA targets. Strong peaks for Rnt1 cleavage (red arrowheads) are detected in (**A**) the known mRNA target *BDF2* and novel targets (**B**) *CAF4*, (**C**) *YDR514C*, and (**D**) *MTM1*. PARE was performed in duplicate, and both independent biological replicates are shown. (**E**) Northern blots detect cleavage products of *BDF2* and *CAF4*. Shown is a

*Figure 2 continued on next page*

*Figure 2 continued*

representative of two independent biological replicates. *PGK1* was used as a loading control. (**F**) Sequence alignment of 42 of 63 mRNA tetraloops and surrounding sequences.

The online version of this article includes the following source data and figure supplement(s) for figure 2:

**Source data 1.** Original files for the northern blots in *Figure 2*.

**Source data 2.** Original files for the northern blots in *Figure 2* with labels of what parts were used in the figure.

**Figure supplement 1.** Examples of novel Rnt1 targets detected by PARE.

**Figure supplement 2.** Gene Ontology analysis of Rnt1 targets.

**Figure supplement 3.** Predicted mRNA tetraloop sequences plus surrounding sequences used for alignment in *Figure 2F*.

pair 4 nts downstream of the cleavage site and destabilizes the *YOR389W* stem loop (ΔG of –15.2 for *YOR389W*, compared to ΔG of –17.8 for *YPL277C*). The hundreds of reads that mapped to *YPL277C* all had the expected G in the fourth position, unambiguously identifying them as *YPL277C* reads. The specificity of *YPL277C* over *YOR389W* confirms previous mutational analysis that sequences in the stem can modulate cleavage (*Lamontagne et al., 2003*; *Lamontagne et al., 2004*).

Because it has been previously suggested that Rnt1 displays a preference for mRNAs involved in carbohydrate metabolism and respiration (*Gagnon et al., 2015*), we investigated whether the Rnt1 mRNA targets identified by PARE function in related pathways. Gene ontology analysis, however, showed no significant enrichment for common cellular processes, molecular function, or cellular localization. Nonetheless, examination of individual targets revealed that 25% of the mRNAs encode transporters or proteins involved in regulating protein targeting to organelles (*Figure 1—figure supplement 2*).

We reasoned that Rnt1 likely selects its mRNA targets by recognizing the same signals used to select ncRNA substrates. Thus, to determine the conservation of the AGNN tetraloop in these mRNA targets, we used sequences surrounding the detected Rnt1 cleavage sites in these ORFs to generate mFold predictions of their secondary structures. Of the 63 ORF sequences, 42 were predicted to fold into double-stranded stem loop structures with A/UGNN tetraloops (*Figure 2—figure supplement 3*). Alignment of these tetraloop sequences revealed a strong preference for A in the first position, with an occasional U (*Figure 2F*; *Figure 2—figure supplement 3*). All aligned sequences possessed a G in the second position of the tetraloop. Although the 2 nts before and after the tetraloop could always form two base pairs, there was no sequence consensus (*Figure 2F*). Similarly, there was no consensus for the nucleotides surrounding the cleavage sites. Taken together, these results indicate that Rnt1 cleavage of mRNAs relies on the presence of a stem loop with an A/UGNN tetraloop sequence (as observed with all known Rnt1 targets) and is influenced by the mRNA secondary structure but not the function or cellular localization of the encoded protein.

## Rnt1 directly and independently cleaves mRNAs

A catalytic mutant of Rnt1 (*rnt1-D245R*) has previously been shown to prevent cleavage in vitro and to be unable to restore growth in a *rnt1Δ* background. This mutant, however, does complement the cell cycle and cell morphology defects of *rnt1Δ*, indicating that Rnt1 has both catalytic and non-catalytic functions (*Catala et al., 2004*). Thus, we used in vivo and in vitro experimental approaches to rule out the possibility that the mRNA cleavage products we detected reflect an indirect effect and/or a non-catalytic function of Rnt1. Specifically, we tested whether the *rnt1-D245R* mutant affected cleavage of mRNAs in vivo. The catalytic mutant was indeed slow-growing and was unable to cleave pre-*SNR83* (*Figure 3A*) like the complete deletion of *RNT1*, confirming previous reports (*Catala et al., 2004*). PARE confirmed that the catalytic mutant resembled *rnt1Δ* in its inability to cleave mRNA targets (*Figure 3B*), consistent with Rnt1 directly cleaving these mRNAs. To complement this analysis and to eliminate the possibility that some cleavage events we detected might be indirect effects, we performed an in vitro cleavage assay in which total RNA isolated from a *rnt1Δ* strain was incubated with varying amounts (0, 4, or 8 pmol) of recombinant Rnt1 purified from *Escherichia coli* (*Figure 3C*). This experimental approach has previously been used to detect Rnt1 cleavage of individual cellular RNAs by northern blot and uses a Rnt1 concentration lower than its in vivo nuclear concentration (see methods). Instead of detecting individually cleaved mRNAs by blotting, we combined this approach with PARE (*Figure 3C*). We found that *BDF2*, *CAF4*, and *YDR514C* are cleaved by Rnt1 in vitro at the

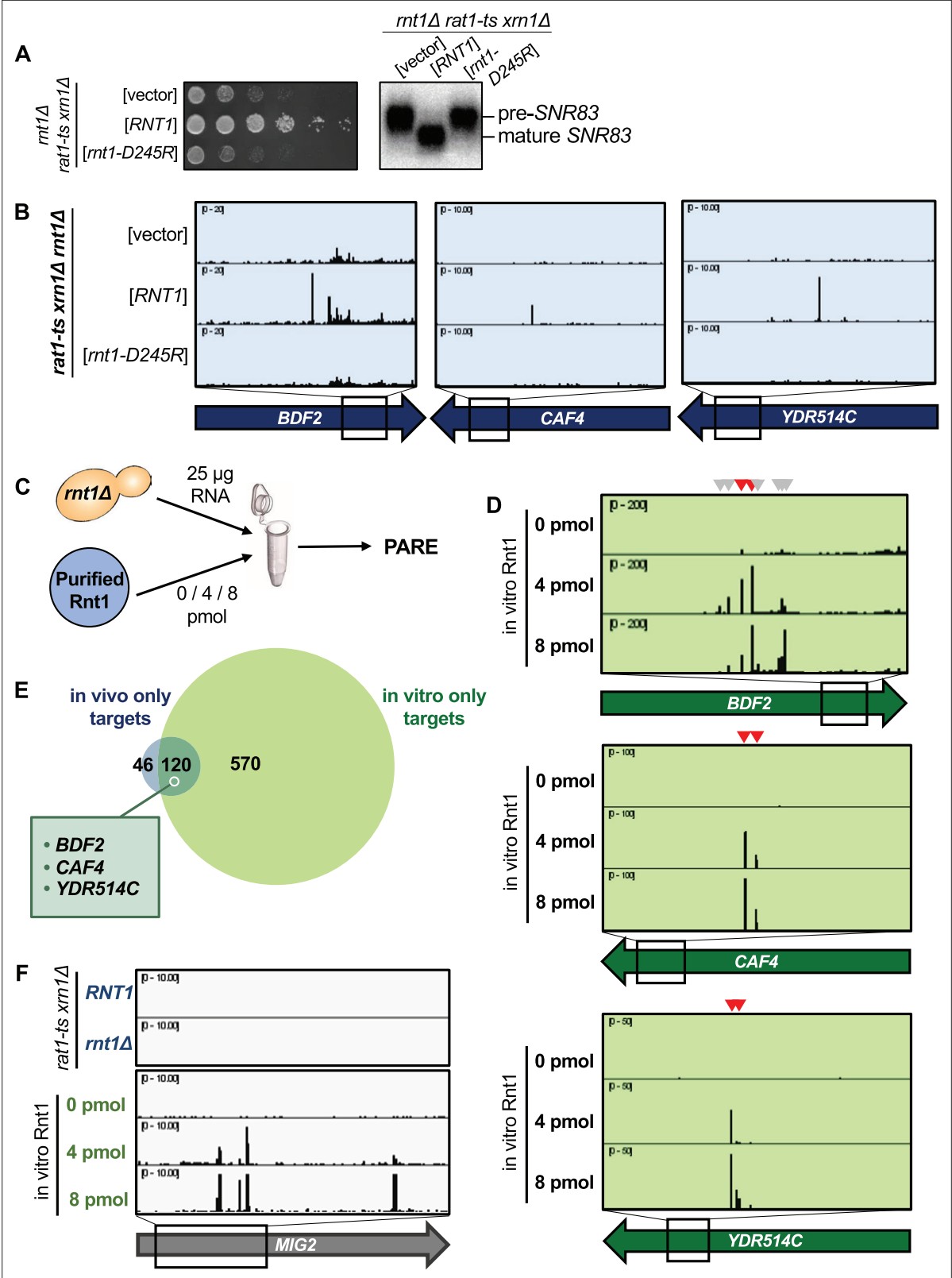

**Figure 3.** Rnt1 directly and independently cleaves mRNAs. (**A**) Validation of Rnt1 catalytic mutant by growth assay and by northern blot of *SNR83*. Growth assay strains were spotted on SC-Leu. Experiment was performed using two independent biological replicates. (**B**) Rnt1 catalytic mutant PARE. Wild-type *RNT1* or *rnt1-D245R* cloned into a plasmid, or empty vector, was expressed in a *rat1-ts xrn1Δ rnt1Δ* triple mutant strain. PARE panels of *BDF2*, *CAF4*, and *YDR514C* are shown. Experiment was performed using three independent biological replicates. (**C**) Schematic of in vitro PARE workflow: RNA

*Figure 3 continued on next page*

*Figure 3 continued*

was isolated from a *rnt1Δ*-only strain and incubated with 0, 4, or 8 pmol of recombinant Rnt1. This RNA was then analyzed by PARE. (**D**) In vitro Rnt1 cleavage of *BDF2*, *CAF4*, and *YDR514C* (red arrowheads, Rnt1 cleavage sites detected in vivo; grey arrowheads, additional Rnt1 cleavage sites detected in vitro, but not in vivo). In vitro PARE was performed using two independent biological replicates. (**E**) Comparison of the numbers of in vivo and in vitro targets. (**F**) In vitro Rnt1 cleavage sites in *MIG2* (panels 3–5) compared to in vivo Rnt1 cleavage sites in *MIG2* (panels 1–2).

The online version of this article includes the following source data for figure 3:

**Source data 1.** Original files for the northern blots in *Figure 3*.

**Source data 2.** Original files for the northern blots in *Figure 3* with labels of what parts were used in the figure.

same sites that are cleaved in vivo (*Figure 3D*), supporting the conclusion that Rnt1 directly and independently cleaves these mRNAs.

Although ~75% of the in vivo targets were also cleaved in vitro, a subset of targets were not detected by our in vitro approach (*Figure 3E*; *Supplementary file 1*). This might be explained by several possibilities. First, the in vitro conditions (ion concentration, pH, temperature, etc.) may not perfectly reflect those of the cell and could potentially interfere with RNA secondary structure, thereby inhibiting cleavage of some targets in vitro. Second, our in vivo PARE analysis (which was performed in a *rat1-ts xrn1Δ* background) required incubation of the strains at 37 °C to inactivate *rat1-ts*, while the in vitro PARE used RNA isolated from a *rnt1Δ* strain grown at 26 °C. Thus, some mRNAs specifically expressed during heat stress would be included in the in vivo dataset but would not be detectable in the in vitro conditions. Third, considering the different strains used for our in vivo versus in vitro PARE experiments (i.e., *rnt1Δ rat1-ts xrn1Δ* vs *rnt1Δ*), there may be mRNAs for which degradation by Rnt1 is redundant with Xrn1- or Rat1-mediated degradation. Fourth, it is possible that Rnt1 may require a cofactor for the cleavage of some specific mRNAs, in which case cleavage would be carried out successfully in vivo, but not in vitro. Indeed, it has previously been shown that the Nop1 protein is required for Rnt1 cleavage of two snoRNAs that lack an obvious AGNN loop (*Giorgi et al., 2001*). Regardless of these possible limitations, our results show that most of the cleavage sites detected by PARE are direct products of Rnt1 and that Rnt1 does not require a cofactor for cleavage of most of its mRNA targets.

Interestingly, there were approximately 7 times as many Rnt1 targets cleaved in vitro than were detectable in vivo (*Figure 3E*). One example is *MIG2* (*Figure 3F*). While our data concur with previous reports that *MIG2* is cleaved by Rnt1 in vitro, cleaved *MIG2* products did not accumulate to exceed our >1 cpm cutoff in the *RNT1* strain (*Figure 1D*). It is possible that these targets are simply expressed at low levels in the cell, potentially during heat stress, and thus would not be detectable by our in vivo experiment. We hypothesized, however, that the drastic increase in Rnt1 targets in vitro could be due to the increased accessibility of these targets to Rnt1, while in vivo, these same targets are mostly cytoplasmic and therefore inaccessible to the primarily nuclear Rnt1.

## Localization is a key determinant in Rnt1 mRNA selection and cleavage

Not surprisingly, colocalization of endoribonucleases with specific mRNAs is critical in determining endoribonuclease target selection in vivo. For example, in the Regulated Ire-Dependent Decay (RIDD) and tRNA Endonuclease-Initiated mRNA Decay (TED) pathways, endoribonucleases on the ER membrane and mitochondrial outer membrane, respectively, degrade mRNAs that are in close proximity due to co-translational protein sorting into those organelles (*Hollien and Weissman, 2006*; *Hurtig et al., 2021*; *Tsuboi et al., 2015*). Rnt1 localization is regulated during the cell cycle, with the enzyme being concentrated in the nucleolus during G1 phase but being more diffusely nuclear during G2/M phases (*Catala et al., 2004*). Additionally, Rnt1 cleavage of *BDF2* is thought to increase during osmotic stress due to *BDF2* retention in the nucleus under this condition (*Wang et al., 2021*). Thus, we hypothesized that some potential in vivo targets might escape Rnt1 cleavage due to their rapid cytoplasmic export.

To examine whether Rnt1 cleavage efficiency of mRNA targets is limited because Rnt1 is sequestered away from the bulk of the mRNA, we performed PARE using Rnt1 mutants that alter its localization within the cell. A *rnt1-K45I* mutant primarily localizes to the nucleolus, while a Rnt1 mutant lacking its nuclear localization sequence (*rnt1-ΔNLS*) localizes to the cytoplasm (*Figure 4A*). Both mutants are fully catalytically active and complement the growth phenotype of *rnt1Δ* (*Catala et al.,*

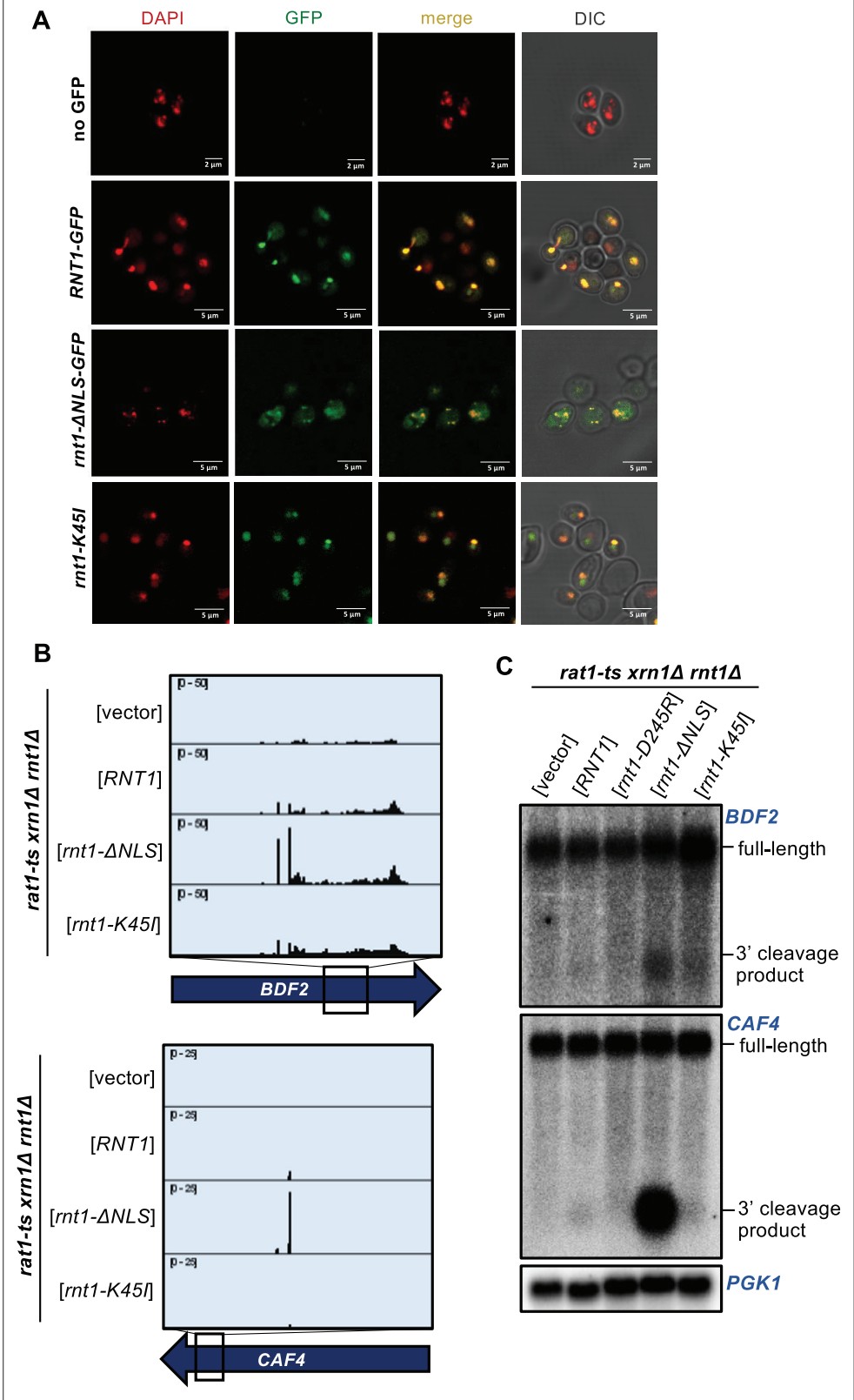

**Figure 4.** Localization is a key determinant in Rnt1 mRNA selection and cleavage. (**A**) Confirmation of Rnt1 cytoplasmic relocalization by confocal fluorescence microscopy. *RNT1-GFP* or *rnt1-ΔNLS-GFP* cloned into a plasmid, or empty vector, was expressed in a *rnt1Δ*-only strain. The nucleus was stained with DAPI, pseudo-colored red. Experiment was performed using two independent biological replicates. (**B**) PARE of *BDF2* and *CAF4* cleaved

*Figure 4 continued on next page*

*Figure 4 continued*

by cytoplasmic Rnt1. Wild-type *RNT1*, *rnt1-ΔNLS-GFP*, or *rnt1-K45I* was expressed in a *rat1-ts xrn1Δ rnt1Δ* triple mutant strain. Experiments were performed using two independent biological replicates. (**C**) Northern blot of *BDF2* and *CAF4* cleaved by cytoplasmic Rnt1. *PGK1* was used as a loading control.

The online version of this article includes the following source data and figure supplement(s) for figure 4:

**Source data 1.** Original files for the northern blots in *Figure 4*.

**Source data 2.** Original files for the northern blots in *Figure 4* with labels of what parts were used in the figure.

**Figure supplement 1.** Effect of Rnt1 localization on cleavage of mRNA and ncRNA targets.

---

*2004*; *Figure 4—figure supplement 1A*). As expected, *rnt1-ΔNLS* cleaved the *BDF2*, *CAF4*, and *YDR514C* mRNAs even more efficiently than wild-type Rnt1 (*Figure 4B and C*; *Figure 4—figure supplement 1B*). The PARE peaks corresponding to the cleaved mRNAs became more pronounced, as did the corresponding bands detected by northern blotting. We did not observe this increase with the more nucleolar *rnt1-K45I*. Instead, we observed increased cleavage of ncRNA sites with nucleolar Rnt1, compared to *RNT1* (*Figure 4—figure supplement 1C*). Notably, although we observed in vitro cleavage of *MIG2*, this target was not detectably cleaved by cytoplasmic Rnt1 (*Figure 4—figure supplement 1D*), indicating that even with increased access to mRNAs, Rnt1 likely does not cleave *MIG2* in an in vivo context. Overall, these results suggest that cleavage of at least some mRNAs is regulated by the localization of Rnt1 and by mRNAs being predominantly cytoplasmic. Conversely, we speculate that Rnt1-mediated mRNA cleavage may be increased under conditions that slow mRNA export from the nucleus or slow Rnt1 import into the nucleolus.

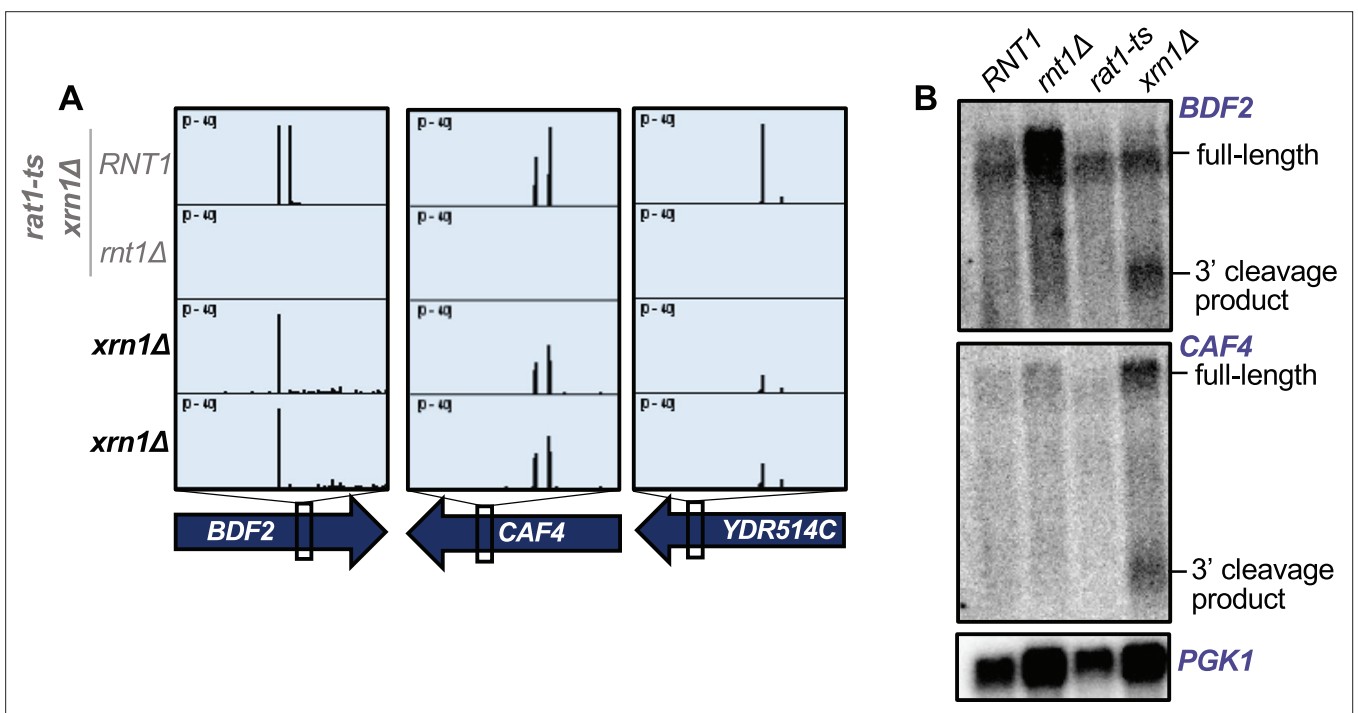

**Figure 5.** Rnt1-cleaved mRNAs are subsequently degraded by Xrn1. (**A**) Rnt1 cleavage peaks in *BDF2*, *CAF4*, and *YDR514C* in *xrn1Δ*-only PARE data. Two independent biological replicates are shown. PARE dataset from the *rat1-ts xrn1Δ* double mutant background is also shown for comparison. (**B**) Northern blot of *BDF2* and *CAF4* using RNA from wild-type *RNT1*, *rnt1Δ*, *rat1-ts*, and *xrn1Δ* single mutant strains. The experiment was performed using two independent biological replicates. *PGK1* was used as a loading control.

The online version of this article includes the following source data for figure 5:

**Source data 1.** Original files for the northern blots in *Figure 5*.

**Source data 2.** Original files for the northern blots in *Figure 5* with labels of what parts were used in the figure.

## Rnt1-cleaved mRNAs are subsequently degraded by Xrn1

To investigate the fate of Rnt1-cleaved mRNAs, we used two complementary approaches. Three different fates have been described for RNAs cleaved by Rnt1. First, pre-snoRNAs are cleaved by Rnt1 upstream of the mature RNA or between two snoRNAs in a polycistronic cluster, which allows for 5′ end maturation by Rat1 (*Qu et al., 1999*). Second, Rnt1 cleaves snRNAs, rRNAs, and some other RNAs co-transcriptionally downstream of their mature RNA ends, which allows for Rat1-mediated degradation of the nascent RNA and transcription termination (*Ghazal et al., 2009*). Third, we have previously shown that if an Rnt1 site is artificially introduced into a reporter mRNA, the 3′ cleavage product is exported to the cytoplasm and degraded by Xrn1 (*Meaux et al., 2011*).

Thus, to determine whether mRNAs are degraded upon Rnt1 cleavage, as well as which enzyme is responsible for this downstream degradation, we examined published *xrn1Δ* PARE data from *Hurtig et al., 2021* and *Hurtig and van Hoof, 2022*. We found that cleaved targets accumulate in the *xrn1Δ* strain, as shown by clear peaks at the Rnt1 cleavage sites in *BDF2*, *CAF4*, and *YDR514C* (*Figure 5A*). This suggests that once these targets are cleaved by Rnt1, they are degraded by Xrn1. To confirm that Xrn1 is the enzyme responsible for degrading these mRNAs following Rnt1 cleavage, we used *rat1-ts* and *xrn1Δ* single mutants to analyze the accumulation of *BDF2* and *CAF4* cleaved transcripts by northern blot. We found that a deletion of *XRN1* led to the accumulation of cleaved *BDF2* and *CAF4* mRNAs, while these cleavage products were still degraded when only Rat1 was inactivated (*Figure 5B*). This indicates that Xrn1 is responsible for the degradation of Rnt1-cleaved mRNA targets and suggests two possibilities: (1) The 3′ cleavage product is exported from the nucleus, allowing for degradation by Xrn1 and avoiding Rat1. The most likely substrate would be newly synthesized RNA, but we cannot exclude that RNA is re-imported into the nucleus for cleavage. (2) These mRNAs escape the nucleus uncleaved and are cleaved by a low amount of cytoplasmic Rnt1. The latter observation, however, is inconsistent with previous findings that the *BDF2* mRNA is cleaved upon nuclear retention (*Wang et al., 2021*). Furthermore, while Rnt1 is synthesized in the cytoplasm, its cytoplasmic concentration is very low (*Figure 4*), presumably because of very efficient nuclear import. Thus, we hypothesize that 3′ cleavage products are exported for Xrn1-mediated decay.

## Rnt1 and decapping products are derived from mRNAs with distinct poly(A) status

Finally, we interrogated the biological relevance of Rnt1 mRNA cleavage. First, we investigated the overall contribution of Rnt1 cleavage to mRNA turnover. The usual methods to measure decay rates depend on the disappearance of the mRNA from a steady-state pool. Because the steady-state pool is predominated by cytoplasmic RNA, these methods are not suitable to assess nuclear degradation. Instead of substrate disappearance, we assessed the abundance of degradation intermediates. The major mRNA degradation pathway is through decapping by Dcp2 (*Muhlrad and Parker, 1992*; *Shyu et al., 1991*), which leads to 5′ monophosphate ends at transcript start sites that can be detected by PARE (*Figure 6*). This allowed us to estimate the contribution of Rnt1 cleavage to the overall mRNA 'degradome' by examining the frequency of Rnt1 cleavage products compared to decapping products (i.e. comparing the number of PARE reads at the transcript start site and at the cleavage site) for each mRNA target. Thus, the relative contribution to decay was estimated as the total number of reads in Rnt1-dependent peaks in a CDS, divided by the total number of reads that reflect both Rnt1 cleavage and decapping. It is important to note, however, that poly(A) shortening usually precedes decapping, so considering that our standard PARE data enriches for poly(A)$^+$ transcripts, it is possible that decapped mRNAs are underrepresented. We therefore focused this analysis on previously published data that used PARE on both oligo(dT)-enriched and oligo(dT)-depleted RNA. Although the PARE library preparation for these samples was slightly different from those we used to identify Rnt1-dependent peaks, we detected the same peaks, indicating the robustness of the analysis. In the poly(A)$^+$ fraction, peaks derived from Rnt1 cleavage predominated over sites derived from decapping (*Figure 6*; *Figure 6—figure supplement 1A*). In contrast, while the Rnt1-dependent sites were still detectable in the poly(A)$^-$ fraction, they were less abundant than decapping sites (*Figure 6*). Thus, we conclude that in contrast to decapping affecting mostly old poly(A)-shortened mRNA, Rnt1 cleaves newly synthesized mRNAs with long poly(A) tails, consistent with its nuclear localization.

We also compared our PARE results to published NET-seq data (*Churchman and Weissman, 2011*). NET-seq maps the 3′ end of RNAs that are still associated with RNA polymerase II by sequencing those

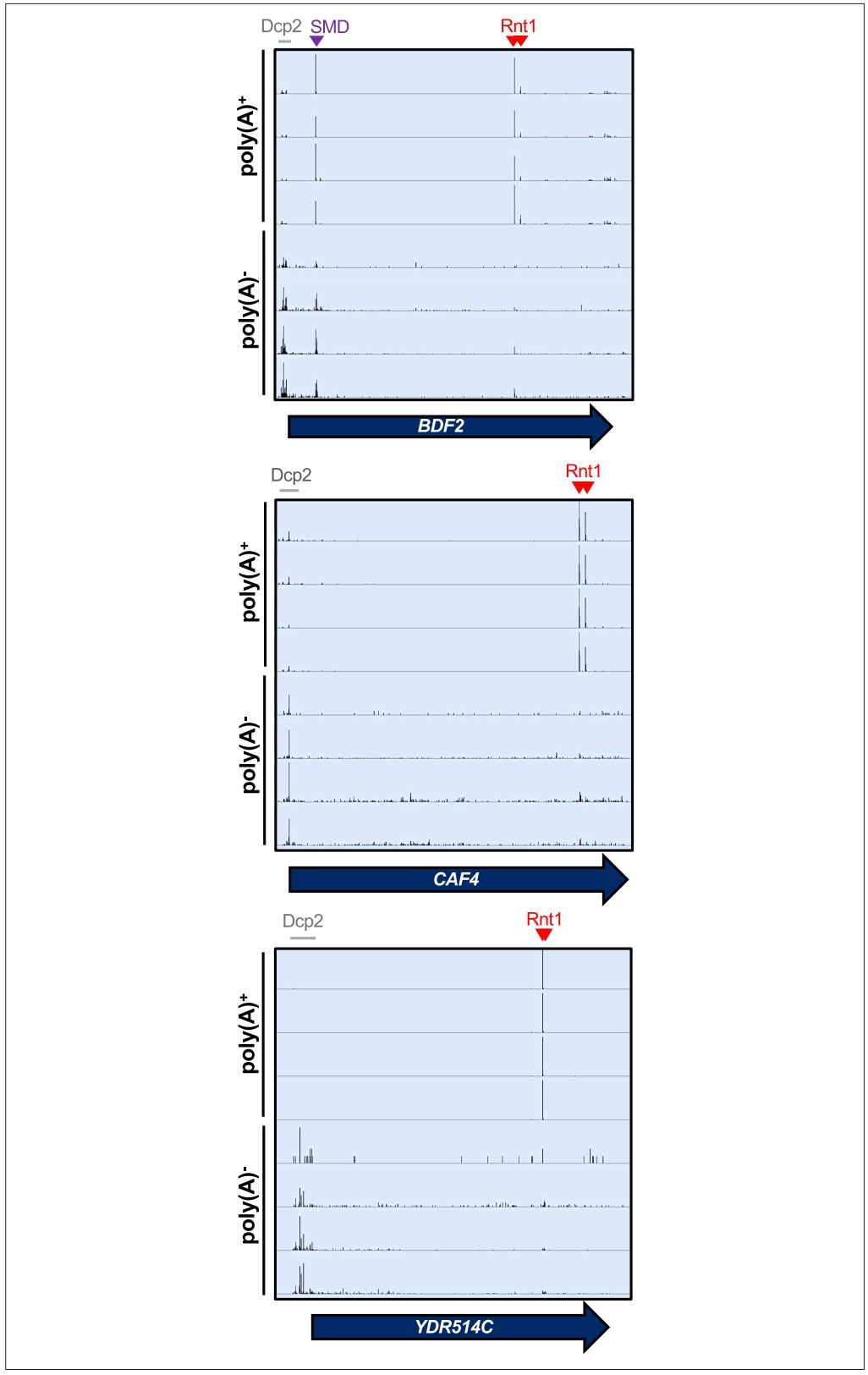

**Figure 6.** Rnt1 and decapping products are derived from mRNAs with distinct poly(A) status. IGV screenshots of *RNT1* vs *rnt1Δ* PARE data generated from poly(A)-enriched and poly(A)-depleted samples show different distributions for Rnt1 products (red arrowheads) and decapping products (grey bars). The *BDF2* mRNA is also a substrate for spliceosome-mediated decay (SMD, purple arrowhead).

*Figure 6 continued on next page*

*Figure 6 continued*

The online version of this article includes the following figure supplement(s) for figure 6:

**Figure supplement 1.** Rnt1 preferentially cleaves polyadenylated mRNA.

**Figure supplement 2.** Rnt1 PARE data generated in this study (top two panels), compared to one replicate of *RNT1* vs *rnt1Δ* RNA-seq data previously published by *Grzechnik et al., 2018* (bottom two panels).

RNAs from the 3′ end. Therefore, co-transcriptional cleavage results in a 3′ end and NET-seq peak that is precisely 1 nt upstream of a 5′ end and PARE peak. As shown in *Figure 6—figure supplement 1B*, PARE and NET-seq identify the expected peaks for the spliceosome-mediated decay (SMD) of *BDF2* (*Volanakis et al., 2013*). In contrast, there were no prominent NET-seq peaks for Rnt1-mediated cleavage of *BDF2*, *CAF4*, or *YDR514C*, although there were sporadic reads for the Rnt1 sites in *BDF2* and *YDR514C*. The absence of prominent NET-seq peaks is consistent with Rnt1 preferentially cleaving after the RNA is polyadenylated and released from RNA polymerase II.

mRNAs with very short poly(A) tails are likely to be translated less efficiently, and thus, decapping of an old mRNA with a short poly(A) tail likely has a smaller effect on gene expression than the cleavage of a newly synthesized mRNA before it can encounter the translation machinery. Thus, the impact of Rnt1 on the expression of mRNAs may be underestimated by the abundance of cleavage products in the degradome.

To determine whether Rnt1 cleavage of mRNAs affects gene expression levels, we re-analyzed a Rnt1 RNA-seq dataset from *Grzechnik et al., 2018*. Unfortunately, the authors performed only a single replicate, and therefore, the data cannot be rigorously analyzed. Nevertheless, we observed that some mRNA targets, such as *CAF4*, *YDR514C*, and *MTM1*, were much more abundant in the *rnt1Δ* sample than the wild-type sample, while others like *BDF2* were unaffected (*Figure 6—figure supplement 2*). This is consistent with Rnt1 cleavage regulating certain mRNAs (like *CAF4*, *YDR514C*, and *MTM1*) but not others, such as *BDF2*, that may undergo more complex regulation to maintain their homeostatic levels.

## Rnt1 cleavage of *YDR514C* mRNA contributes to normal cell growth

As a complementary approach to interrogate the biological relevance of Rnt1 mRNA cleavage, we identified spontaneous suppressors that restored growth to a *rnt1Δ* strain. To isolate suppressors, we grew 13 cultures of a *rnt1Δ* strain to saturation, diluted them 1000-fold, and repeated the growth for 10 cycles (*Figure 7A*). Each of the final cultures grew substantially faster than the starting strain (*Figure 7—figure supplement 1*). We then isolated an individual colony from the final culture and sequenced its genome (*Figure 7A*). Most of the mutations we identified were in *RRP6*, *RRP47*, and *MTR4*, which encode cofactors of the nuclear RNA exosome (*Figure 7—figure supplement 1*). This suppression probably reflects the fact that Rnt1 and the RNA exosome are both involved in processing rRNA, snRNA, and snoRNA precursors (*Allmang et al., 1999*; *Chanfreau et al., 1997*; *Abou Elela and Ares, 1998*; *Seipelt et al., 1999*) but is not informative as to which of these RNA processing defects causes the slow growth of *rnt1Δ*.

Our most interesting mutation, however, was in the newly identified Rnt1 target *YDR514C* (*Figure 7—figure supplement 1B*). *YDR514C* encodes a putative nuclease of unknown function, and our spontaneous suppressor strain (evolved strain 13; *Figure 7A*) contained a G220S mutation in the nuclease domain (*Figure 7—figure supplement 1B*). This mutation is 420 nts upstream of the most proximal Rnt1 cleavage site. Of note, this glycine is 100% conserved in the Saccharomyceta-ceae (budding yeast) family. The most likely effect of this mutation is that it disturbs the function of the Ydr514c protein. Although Rnt1 cleaves 420 nts downstream of this mutation, cleavage is still within the nuclease domain. Therefore, Rnt1 cleavage of *YDR514C* and the G220S mutation are both expected to reduce Ydr514c function, providing an explanation for how the G220S mutation might compensate for the absence of Rnt1.

While most of the spontaneous suppressor strains contained a single mutation, the evolved strain 13 contained a second mutation in *PUF4*. To determine whether loss of *YDR514C* function contributed to the suppression we observed in evolved strain 13, we constructed a *rnt1Δ ydr514cΔ* double mutant. Interestingly, we observed improved growth with this strain compared to *rnt1Δ*, indicating that *YDR514C* loss of function suppresses the *rnt1Δ* growth defect independently of the *PUF4* mutation

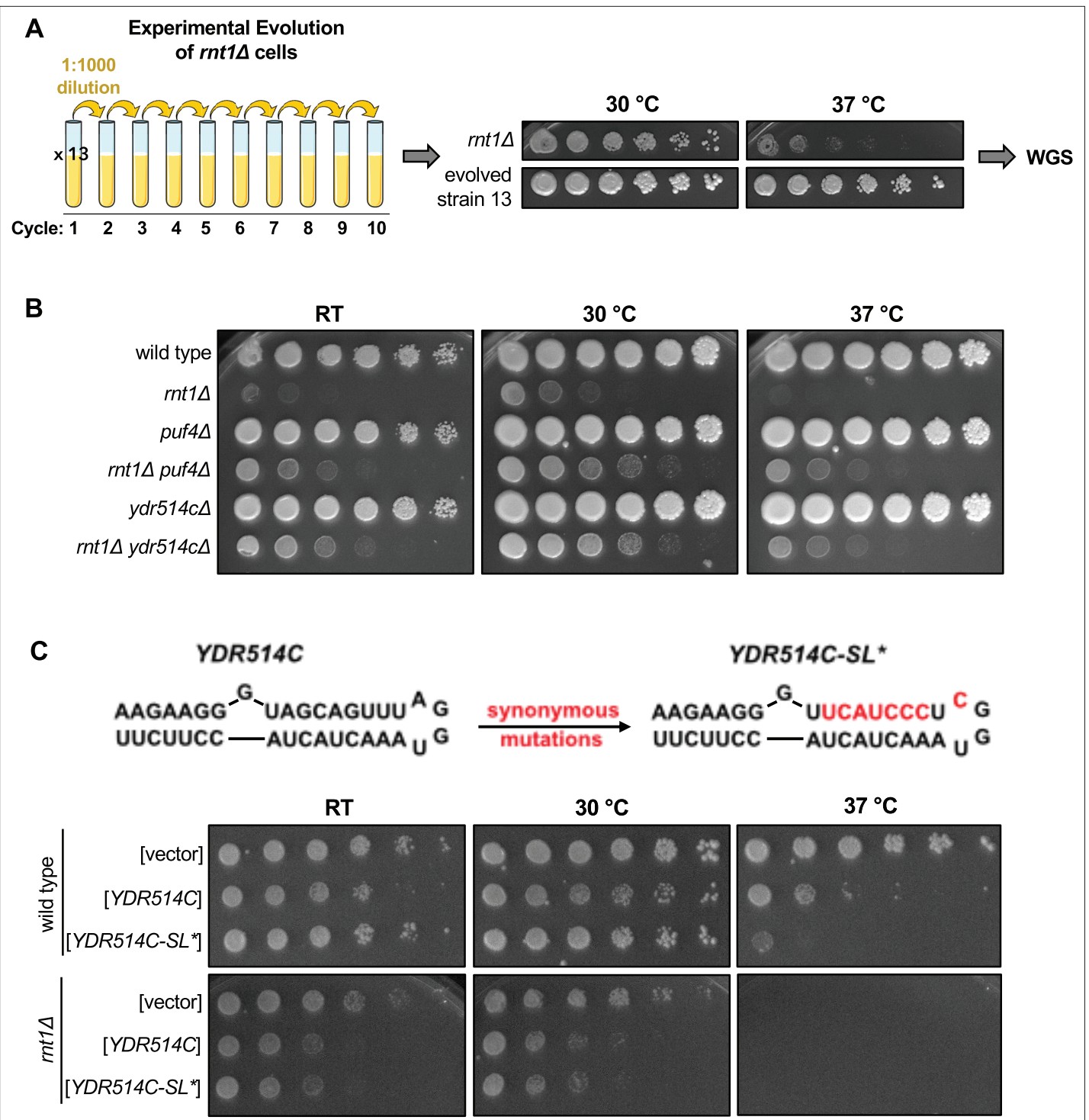

**Figure 7.** Rnt1 cleavage of *YDR514C* mRNA contributes to normal cell growth. (**A**) Schematic of *rnt1Δ* experimental evolution. Thirteen cultures of *rnt1Δ* were grown to saturation, then sub-cultured to a 1:1000 dilution for 10 cycles. Solid media growth assays were performed, and strains showing enhanced growth compared to the *rnt1Δ* parent strain were analyzed by whole-genome sequencing (WGS). The growth assay of evolved strain 13 is depicted above. (**B**) Growth assay confirming enhanced growth of a *rnt1Δ ydr514cΔ* double mutant compared to *rnt1Δ*. Experiment was performed using two independent biological replicates. (**C**) Growth assay confirming impaired growth of wild type and *rnt1Δ* strains harboring plasmids that overexpress wild-type *YDR514C* or the *ydr514c* stem loop mutant. Experiment was performed using two independent biological replicates. Strains were spotted on SC-Leu.

The online version of this article includes the following figure supplement(s) for figure 7:

**Figure supplement 1.** Isolation of suppressors of rnt1Δ.

(*Figure 7B*). This suggests to us that a lack of *YDR514C* cleavage may contribute to the *rnt1Δ* slow-growth phenotype and that this genetic interaction plays a role in maintaining normal cell growth.

To further substantiate that Rnt1 cleavage of *YDR514C* contributes to maintaining cellular homeostasis, we investigated whether overexpression of *YDR514C* results in slow growth. In this experiment, either wild-type or a stem loop mutant of *YDR514C* (*YDR514C-SL\**) was cloned into an overexpression vector. The *YDR514C-SL\** mutant changes three codons to maintain the wild-type amino acid sequence (Ser-Ser-Leu) but disrupts the predicted stem loop (*Figure 7C*). We noted two interesting results. First, *YDR514C* overexpression is toxic in the *rnt1Δ* strain, but less so in the *RNT1* control strain. This is consistent with suppressor 13 being selected because of the *ydr514c* mutation. Second, in the *RNT1* strain, *YDR514C-SL\** was more toxic at 37 °C than wild-type *YDR514C*, but both plasmids behave identically in the *rnt1Δ* strain. This is consistent with our hypothesis that Rnt1 cleavage of *YDR514C* mRNA limits its toxicity. Thus, we propose that Rnt1 cleavage of *YDR514C* may prevent the aberrant expression of this mRNA, which contributes to maintaining cellular homeostasis. Together, these results reveal biologically relevant roles for Rnt1 mRNA cleavage in regulating the turnover and/or gene expression levels of specific mRNAs and in maintaining normal cell growth through the cleavage of the *YDR514C* mRNA.

## Discussion
### Expanding the Rnt1 target repertoire

Here, we greatly expand the known mRNA target repertoire of Rnt1 and highlight the biological importance of its mRNA cleavage function. Using PARE, we identify a novel set of 60 mRNAs that are cleaved by Rnt1 in vivo. As with ncRNA targets, we find that Rnt1 cleaves mRNAs that possess double-stranded stem loops with terminal AGNN tetraloops. Using a Rnt1 catalytic mutant to perform PARE, we further show that the catalytic activity of Rnt1 is required for mRNA cleavage. Although this study does not experimentally determine direct Rnt1 binding sites in mRNA targets, the cleavage sites have the hallmarks of direct cleavage by Rnt1: The sites are at –14 and +16 of AGNN tetraloops and produce a typical 3″ 2-nt overhang. Furthermore, the cleavage sites detected by PARE in vivo are not observed when the catalytic capability of Rnt1 is abrogated. We also find that Rnt1 is able to cleave most mRNAs in vitro at sites identical to those cleaved in vivo. All these observations indicate direct cleavage of these targets by Rnt1. Importantly, this includes the mRNA target *YDR514C* – cleavage of which we found to be critical for maintaining normal cell growth. The in vitro PARE results further demonstrate that Rnt1 does not require a cofactor for cleavage of most mRNAs. For the small subset of mRNAs that were not cleaved in vitro, further analysis is needed to determine whether these targets fold differently in vitro; are only expressed (or fold differently) during heat stress; can be degraded by Rat1 or Xrn1 without Rnt1-specific cleavage; and/or require a cofactor for Rnt1 to cleave.

Rnt1-cleaved mRNAs encode proteins with no common biological function or subcellular localization. However, the localization of the mRNA targets to the cytoplasm, in contrast to Rnt1's nuclear localization, limits cleavage of these targets. This partially explains why Rnt1 cleaves only specific mRNAs in vivo, despite its capacity to cleave a wider set of mRNAs in vitro. One outstanding question is whether mRNAs efficiently cleaved by Rnt1 remain in the nucleus for longer periods, perhaps due to slower export rates, compared to mRNAs that are less efficiently cleaved.

Once these mRNA targets are cleaved by Rnt1, they are degraded by Xrn1, not by Rat1. This likely reflects that Rnt1-cleaved mRNAs are exported to the cytoplasm for further degradation by Xrn1. In support of the alternate possibility that some mRNAs can be cleaved in the cytoplasm, we show that Rnt1 relocalized to the cytoplasm retains its ability to cleave mRNAs. This would also present the most energetically economic scenario for the cell. However, Rnt1 cytoplasmic localization conflicts with the scarcely detectable level of Rnt1 in the cytoplasm and with our current understanding of the localization of this well-studied nuclease. It has also been shown that *BDF2* cleavage increases upon nuclear retention (*Wang et al., 2021*) and that the 5′ cleavage product of *BDF2* accumulates in the absence of the nuclear exosome subunit Rrp6 (*Roy and Chanfreau, 2014*). Neither of these would be expected if *BDF2* were cleaved by a pool of cytoplasmic Rnt1. Nuclear cleavage by Rnt1 followed by export of the 3′ cleavage product to the cytoplasm seems most consistent with all the available data.

This study has also expanded the number of known Rnt1 targets in other classes of RNAs, specifically UTRs of mRNAs, pre-mRNA introns lacking snoRNAs, antisense transcripts, and regions

annotated as intergenic. These sites may be cleaved to initiate degradation or a processing event, and cleavage may be essential for the regulation of these targets or may be redundant with other pathways. Of the intronic sites, five did not contain snoRNAs. Most previously reported Rnt1-cleaved introns encode snoRNAs, with exceptions being the spliced lariat intron of *RPL18A*, which is degraded upon Rnt1 cleavage, and the first intron of *RPS22B*, which is cleaved to initiate the degradation of unspliced *RPS22B* transcripts (*Danin-Kreiselman et al., 2003*). Thus, these five newly identified Rnt1-cleaved introns lacking snoRNAs dramatically expand this class of targets. It remains to be determined whether these introns are cleaved to trigger the degradation of unspliced transcripts or whether cleavage and degradation occur after splicing. Also of note, the intergenic regions targeted by Rnt1 included three that contain uncharacterized ncRNAs (*Gao et al., 2021*; *Nagalakshmi et al., 2008*), possibly representing a novel class of Rnt1 targets. These sites may be cleaved to liberate the ncRNAs, while other intergenic sites may be cleaved to prevent transcriptional read-through, as in the case of the *NPL3-GPI17* dicistronic transcript (*Ghazal et al., 2009*).

## Biological relevance of Rnt1 mRNA cleavage

There are diverse implications of Rnt1 mRNA cleavage. First, Rnt1 appears to recognize the same secondary structures for mRNA degradation and ncRNA processing. For seven targets in particular, almost all of the poly(A)$^+$ degradation products we detected reflect Rnt1-initiated degradation, and each of them had an obvious, strong predicted stem-loop structure. For others, Rnt1 products formed a smaller fraction of the degradome, and a number of these mRNAs did not have an obvious, strong predicted stem-loop structure. We suspect that they may form weak stem loops and/or may adopt multiple alternative structures in vivo. Specific RNA binding proteins may also affect cleavage by affecting the mRNA structure, analogous to Nop1's effect on cleavage of the *SNR18* and *SNR38* snoRNAs (*Giorgi et al., 2001*). Additionally, other factors, including the sequences surrounding the AGNN loop and those around the cleavage site, affect Rnt1 cleavage efficiency (*Lamontagne et al., 2003*; *Lamontagne et al., 2004*). The example of *YPL277C* indeed highlights the importance of the stem for target selection, as a single base difference from *YOR389W* in the stem sequence affects cleavage by Rnt1. Another notable target we identified was *MTM1*, which did not form the typical extended stem-loop recognized by Rnt1. We suspect that its unique structure results in cleavage at a single site. Overall, Rnt1 cleavage appears to be finely tuned, and the targets implicate the enzyme in regulating a range of biological pathways, including protein targeting and membrane assembly, DNA damage response, and various chemical biosynthesis pathways. In future studies, it will be interesting to dissect the biological consequences of Rnt1 cleavage of individual mRNA targets beyond *YDR514C* – particularly those that are also cleaved by Rnt1 in vitro.

Second, the mRNA pools that are targeted by Rnt1 and Dcp2 appear to have distinct polyadenylation states. Rnt1 products are relatively more abundant in the oligo(dT)-selected pool, while decapping products are relatively more abundant in the oligo(dT)-depleted pool. We conclude that Rnt1 preferentially cleaves newly made mRNAs that have not had their poly(A) tail shortened significantly, while Dcp2 preferentially cleaves old mRNAs that have undergone poly(A) tail shortening. This suggests that Rnt1 cleavage is not used to reduce the mRNA half-life of productive cytoplasmic mRNAs. Overall, our results suggest that a fraction of mRNA is cleaved before export. Competing alternative fates – cleavage versus export – form a kinetic proofreading mechanism that appears well-suited for mRNA quality control: proper mRNAs that are rapidly exported escape Rnt1, while aberrant mRNAs that are slowly exported are degraded. Our observation that many more mRNAs can be cleaved by Rnt1 in vitro suggests that they are subject to this kinetic proofreading. The observation that many of them are not detectably cleaved in vivo can be explained by them passing this quality control check. The aberrancies that reduce mRNA export and trigger this quality control mechanism are likely to differ for the mRNAs we identified here, and mRNA export could be regulated by internal and environmental signals such that under other conditions another subset of mRNAs is cleaved.

Third, the genetic interaction between *RNT1* and *YDR514C* suggests to us that Rnt1-mediated cleavage of the *YDR514C* mRNA may have important biological consequences. The function of Ydr514c is unknown, but we postulate that its expression needs to be tightly controlled. We hypothesize that *rnt1Δ* results in aberrant *YDR514C* expression, which is detrimental to growth. This explains why a *rnt1Δ* strain is slow-growing, but a deletion of *YDR514C* partially suppresses the slow-growth phenotype. It does not appear likely that the spontaneous *YDR514C* G220S mutation interferes with

Rnt1 cleavage. Rather, we suspect that the mutation results in loss of function of the encoded protein, and therefore reduced toxicity to the cell. Indeed, overexpressing *YDR514C* led to a considerable growth defect, while disrupting the *YDR514C* stem-loop (thereby inhibiting Rnt1 cleavage) further delayed growth only in the presence of Rnt1. Uncovering the function of *YDR514C* will provide clearer insights into its mechanism of suppression, which will be addressed in future studies.

The lack of complete suppression by *ydr415cΔ*, however, indicates that this is not the only detrimental effect of *rnt1Δ*. We identified mutations in *PUF4* and genes encoding the nuclear RNA exosome cofactors Rrp6, Rrp47, and Mtr4 as other suppressors of *rnt1Δ*. Puf4 is an mRNA binding protein that regulates the stability of hundreds of mRNAs that are required for ribosome biogenesis, and Rrp6, Rrp47, and Mtr4 are all required for rRNA maturation. This suggests that Rnt1 indeed plays a role in ribosome biogenesis that affects growth, and we can envision three possible mechanisms for this. First, Rnt1 plays a direct role in pre-rRNA processing. Second, Rnt1 has a function in the maturation of pre-snRNAs that are critical for splicing, and introns are enriched in ribosomal genes. Third, Rnt1 and the RNA exosome cofactors are both required for snoRNA processing, which in turn is required for ribosome biogenesis. Strikingly, the snoRNP subunits Nop1, Nop56, Nop58, Gar1, Nhp2, Nop10, and Cbf5 have all been identified as targets of Puf4 (*Lapointe et al., 2017*). Deletion of most individual snoRNAs has no effect on growth, but a few are essential (e.g. U3 snoRNA). Thus, defects in processing one of these essential snoRNAs or the simultaneous defect in processing multiple snoRNAs may explain the *rnt1Δ* growth defect and its suppression by RNA exosome and *puf4* mutations.

While our results suggest that Rnt1 affects growth through cleavage of the *YDR514C* mRNA under lab conditions, the other identified Rnt1 mRNA targets affect a range of cellular processes, and may affect growth under specific cellular conditions. Furthermore, the broad distribution of Rnt1 cleavage sites throughout the yeast transcriptome in regions other than mRNA CDSs and ncRNAs (i.e. 3' and 5' UTRs, intergenic regions, introns, and antisense transcripts) suggests to us that the cleavage function of Rnt1 may exert its effect on multiple cellular processes and pathways not yet discovered for the enzyme. An additional possibility is that Rnt1 is important for the eradication of transcripts that are poorly exported from the nucleus and/or under conditions in which export is blocked. In this role, Rnt1 would act as a quality control on nuclear export competence similar to the role of nonsense-mediated mRNA decay in the quality control of translation. Our finding that most mRNA targets contain sites that can be cleaved in vitro suggests that such a quality control mechanism could survey many mRNAs.

## Implications for human RNase III

While the details of RNA recognition vary, the fundamentals of Rnt1 function resemble those of other RNase III family members. This includes the human homolog Drosha, and Dcr1 in many fungi (but not *S. cerevisiae*; *Court et al., 2013*; *Nicholson, 2014*). As RNase III family enzymes, they cleave double-stranded stem loops and leave 2-nt 3' overhangs (*Court et al., 2013*; *Provost et al., 2002*). Drosha and *S. pombe* Dcr1 are also nuclear enzymes with prominent roles in the processing of ncRNAs (*Court et al., 2013*; *Emmerth et al., 2010*; *Provost et al., 2002*), and Drosha has previously been described to cleave a few select mRNAs (*Han et al., 2009*; *Karginov et al., 2010*). In contrast, *S. castellii* Dcr1 has been reported to be cytoplasmic (*Szachnowski et al., 2019*) and thus improperly localized for surveillance of nuclear export competence. Although Rnt1, Dcr1, and Drosha differ in their ncRNA targets (rRNA, snRNA, and snoRNA vs miRNA) and in the way they recognize substrates, each enzyme utilizes the same mechanism to cleave its respective ncRNA and mRNA targets (*Court et al., 2013*; *Nicholson, 2014*).

The ability of Drosha and Rnt1 to cleave mRNAs suggests that eukaryotic RNase III family members are multifunctional and active in mRNA degradation, in addition to their earlier discovered ncRNA processing functions. We speculate that the mRNA cleavage role of Rnt1 has been more difficult to detect because only a fraction of each individual mRNA is cleaved, and such nuclear cleavage of a subset of RNAs is not readily detectable by methods that measure mRNA stability by disappearance of a steady-state pool. This limitation would equally apply to Drosha, and therefore it appears likely that Drosha may also play a more prominent role in mRNA degradation than has been previously described. It would be interesting to see whether PARE could detect a more extensive effect of Drosha cleavage on the human protein-coding transcriptome. Performing this in a genetic background devoid of *XRN1* and the Rat1 ortholog *XRN2* makes such PARE experiments challenging, but our results imply that an *xrn1Δ* background may be sufficient to identify at least some Drosha targets.

A recent study has uncovered two independent missense variants in *DROSHA* as a cause of neuro-developmental disease (including severe intellectual disability, white matter atrophy, microcephaly, epilepsy, and dysmorphic features; *Barish et al., 2022*), making the need for a full understanding of Drosha function even more important. More broadly, the quality control of mRNA export competence that we propose results from Rnt1 cleavage does not require any specific feature of Rnt1 other than its nuclear localization. It is therefore possible that other nuclear RNases play a similar role. Indeed, a similar role has been proposed for the yeast endoribonuclease Swt1, and PARE analysis could be used to identify its targets.

## Applications and advantages of PARE

As mentioned, PARE can be used for the detection of non-classical targets of other well-known endoribonucleases. Cellular conditions can also be modulated to determine the specific circumstances under which these enzymes might cleave atypical classes of substrates. Additionally, this technique might be applied to the study of poorly characterized proteins that appear to be endoribonucleases. This may include proteins that possess a predicted nuclease domain, interact with other RNA processing or decay factors, and/or localize to sites of RNA processing or degradation in the cell.

Combining the current characterization of Rnt1 with our previous analysis of TSEN reveals some informative commonalities. Rnt1 and TSEN both have very well-characterized functions in the processing of stable ncRNAs, and inactivating either one of these enzymes causes a large reduction in growth rate and many downstream effects. This makes it challenging to find direct RNA substrates through traditional RNA sequencing. In contrast, by sequencing RNA cleavage products, PARE can capture the primary products of the cleavage events as they have occurred in the cell. In the case of TSEN, we showed that its inactivation induced the general stress response, and the large number of genes affected by the general stress response made the direct TSEN targets undetectable by RNAseq. PARE, however, readily detected TSEN-cleaved mRNAs. Similarly, several previous studies have identified many effects of Rnt1 but failed to detect prominent mRNA cleavage events (other than *BDF2*). We suspect that there are other RNases well-characterized for other functions that may moonlight as mRNA degradation enzymes.

## Materials and methods
### Yeast strains and plasmids

All yeast strains and plasmids used in this study are listed in *Supplementary files 2 and 3*, respectively. The *rnt1Δ::NEO*, *puf4Δ::NEO*, and *ydr514cΔ::NEO* single mutants were obtained from the MATa haploid yeast gene knockout collection (here). The *rat1-ts::URA3* strain was obtained from the Hieter essential yeast gene temperature-sensitive collection (*Kofoed et al., 2015*). The *xrn1Δ::HYG* strain was created by *Hurtig et al., 2021*. The *rnt1Δ::HYG* strain was created by standard cloning methods. Briefly, the pAG32 vector containing the hygromycin resistance (*HYG^R*) gene cassette was cut with HindIII-HF (New England BioLabs, cat. R3104S, USA) and SpeI-HF (New England BioLabs, cat. R3133S, USA) restriction enzymes, and vector fragments were separated on a 1% agarose gel. The *HYG^R* cassette (~1.75 kb) was purified using the Zymoclean Gel DNA Recovery Kit (Zymo Research, cat. D4008) and transformed into the *rnt1Δ::NEO* strain using standard LiAc transformation methods. Positive transformants were selected on YPD +hygromycin plates and counter-selected on YPD +neomycin plates. All other strains were created by standard genetic crosses and LiAc transformations. Overexpression plasmids containing wild-type *YDR514C* or the *YDR514S-SL\** mutant were generated by Gibson assembly using oligos listed in *Supplementary file 4*.

### Yeast growth conditions

All strains used for in vivo PARE experiments were grown at room temperature, then shifted to 37 °C for 1 hr to inactivate *rat1-ts*. The *rnt1Δ::HIS3* strain used for in vitro PARE, *rnt1Δ::NEO* strains used for fluorescence microscopy, and all strains used for experimental evolution were grown at 30 °C. For solid-media growth assays, temperature-sensitive strains were grown at 27 °C, and all other strains at 30 °C. Liquid cultures of strains containing plasmids were grown in SC-Leu, and all other strains were grown in YPD. All solid-media growth assays were spotted on YPD, unless otherwise noted.

## Solid media growth assays

Yeast cultures were grown to mid-log phase (OD$_{600}$ of ~0.6) at room temperature for temperature-sensitive strains, or at 30 °C for all other strains. Cells were washed, resuspended to an OD$_{600}$ of 0.6, then serially diluted and spotted on YPD or SC-Leu.

## RNA isolation

For in vivo PARE experiments and northern blots, yeast cells were grown in 20 mL cultures to mid-log phase (OD$_{600}$ of 0.6–0.8) at 27 °C, then shifted to 37 °C for 1 hr to inactivate *rat1-ts*. Cells were harvested by centrifugation, and RNA was isolated using hot phenol as previously described by *He and Jacobson, 1995*. In brief, cells were resuspended in 500 µL cold RNA buffer A (3 M NaOAc, 0.5 M EDTA pH 8.0) and 500 µL phenol heated to 65 °C, then incubated at 65 °C for 4 min, with vortexing for 10 s/min. The cell suspension was then centrifuged and the aqueous layer collected and mixed with 500 µL phenol heated to 65 °C. The incubation, vortexing, centrifugation, and aqueous layer collection steps were repeated as described above. The aqueous layer was then mixed with phenol:chloroform (1:1), centrifuged, and removed to a new tube. Total RNA was precipitated overnight in 100% ethanol at –80 °C. For in vitro PARE, the *rnt1Δ::HIS3* strain was grown to mid-log phase (OD$_{600}$ of 0.4–0.6) at 26 °C and a 50 mL culture was harvested for total RNA isolation as described by *Catala and Abou Elela, 2019*.

## In vitro cleavage assay

25 µg total RNA isolated from *rnt1Δ::HIS3* was incubated with either 0, 4, or 8 pmol of recombinant Rnt1 purified from *E. coli* in 100 µL reaction buffer (30 mM Tris pH 7.5, 150 mM KCl, 5 mM spermidine, 0.1 mM DTT, 0.1 mM EDTA) as described by *Catala and Abou Elela, 2019*. MgCl$_2$ was added to a final concentration of 10 mM to initiate the cleavage reaction, which was allowed to proceed for 20 min at 30 °C. The reaction was stopped by adding 1 volume LETS buffer (100 mM LiCl, 10 mM EDTA, 10 mM Tris, 0.2% SDS pH 7.5), and the cleaved RNA was isolated using phenol:chloroform:isoamyl alcohol (25:24:1). The isolated RNA was then submitted to LC Sciences for PARE.

The abundance of Rnt1 has been estimated to be ~2,512 molecules/cell (*Ho et al., 2018*). Given that the yeast cell nucleolus is about 0.6 fL, if 2,512 molecules of Rnt1 were uniformly distributed through the nucleolus its concentration would be 7 µM (2,512 molecules/nucleolus in 0.6 fL) (*Uchida et al., 2011*), or approximately 100 times higher than our highest in vitro concentration (80 nM).

## Parallel analysis of RNA ends (PARE)

PARE was performed as previously described by *Hurtig et al., 2021*. 10 µg of RNA isolated from each strain, or 10 µg of RNA cleaved by recombinant Rnt1, was used for two rounds of poly(A)$^+$ RNA enrichment. T4 RNA ligase was used to ligate 5' adapters onto the 5'-monophosphate ends of 3' cleavage products. NEBNext Ultra II RNA Library Prep Kit (New England BioLabs, cat. E7770, USA) was used for cDNA library preparation: Reverse transcription was performed using a 3'-adapter random primer for first-strand cDNA synthesis, and the cDNAs were amplified by PCR (3 min at 95 °C; 15 s at 98 °C for 15 cycles; 15 s at 60 °C, 30 s at 72 °C; and 5 min at 72 °C). AMPureXP (Beckman Coulter, ref A63882, USA) beads were used for size selection (200–400 bp), and 50 bp single-end sequencing was performed with Illumina Hiseq 2500. The PARE libraries to compare poly(A)-enriched to poly(A)-depleted RNAs from Hurtig et al. were generated in the lab of Pamela Green according to their protocol, which is slightly different. The same peaks were identified, indicating the robustness of the data.

Raw PARE (and NET-seq) data files were uploaded onto the Galaxy online server (usegalaxy. org), and the following tools were used with default parameters, unless otherwise noted, to analyze sequencing data (*Figure 1—figure supplement 1*). The sequencing reads were aligned to the R64-1-1 *Saccharomyces cerevisiae* reference genome using TopHat (*Trapnell et al., 2009*; with parameters library type: FR First Strand, minimum intron length: 30, maximum intron length: 5000). For each sample, the resulting .bam file was used to count the number of reads starting at every position in the genome, for both forward and reverse strands, using the bamCoverage tool (*Ramírez et al., 2014*) to generate both bigwig and bedgraph files (with parameters bin size: 1, scaling/normalization: cpm, coverage file format: bigwig/bedgraph, include reads originating from fragments: forward/reverse). The bigwig and bedgraph files store the PARE score values for individual samples. The bigwigCompare tool (*Cohen et al., 2016*) was then used to compare the number of reads between samples, for

each strand, using the bigwig files generated from bamCoverage (with parameters how to compare: log2 of the signal ratio, pseudocount: 0.01, coverage file format: bigwig/bedgraph, length in bases of the non-overlapping bins: 1). In this step, the comPARE score or modified log2(fold change) is calculated as the ratio of read coverage between samples. To account for the lack of cleavage at Rnt1 sites expected in the *rnt1Δ* strain, a 'pseudocount' of 0.01 is used to avoid dividing by zero. For each strand, the bedtools Merge BedGraph files tool (*Quinlan and Hall, 2010*) was used to combine the bedgraph files generated by bamCoverage and bigwigCompare (with parameter report empty regions: no). The resulting merged bedgraph file contains the PARE score (peak height) for each sample and the comPARE score (change in peak height in *rnt1*) for each *rnt1* sample. The Filter tool was used to filter rows (positions in the genome) in the merged bedgraphs to include only positions with reads >1 cpm in the *RNT1* strain (*RNT1* column 'c'>1). Filtered merged bedgraphs were exported for further analysis in Microsoft Excel, and bigwig files generated from bamCoverage and bigwigCompare were exported for visual analysis and figure generation in the Integrative Genomics Viewer (IGV).

A list of all published Rnt1 cleavage sites was compiled by mining the literature, and a BED file of these sites (which includes the chromosome, strand, and start and end positions of cleavage) was created. This BED file was analyzed alongside the PARE data generated in this study.

## Whole-genome sequencing

The *rnt1Δ::NEO* parent strain and evolved strains were submitted to SeqCenter for whole-genome sequencing. The returned raw sequencing reads were trimmed using Trim Galore! (*Krueger, 2015*), quality control was performed using FastQC (*Andrews, 2010*), and reads were mapped to the yeast genome using Bowtie2 (*Langmead and Salzberg, 2012*). FreeBayes (*Garrison and Marth, 2012*) was used to detect variants (with parameters Require at least this coverage to process a site: 20, ploidy: 1, Report sites if the probability that there is a polymorphism at the site is greater than: 0.5). Candidate mutations were visualized in IGV, and VEP (https://useast.ensembl.org/Tools/VEP) was used to annotate the effects of the identified mutations, including affected genes.

## Northern blots

Northern blots of *SNR83* were performed by separating 10 µg of total RNA from the indicated strains on a 1.3% denaturing agarose formaldehyde gel in 1 X MOPS buffer (MOPS, NaOH to pH 7.0, 3 M NaAc, 0.5 M EDTA pH 8). RNA samples were resuspended in 10 µL formaldehyde loading dye (Invitrogen, ref 8552, USA) prior to loading. Northern blots of *BDF2* and *CAF4* were performed by separating 7 µg of poly(A)-enriched RNA in formaldehyde loading dye on 1.3% denaturing agarose gels. Poly(A)$^+$ RNA enrichment was performed using Poly(A)Purist-MAG Magnetic mRNA Purification Kit (Invitrogen, ref AM1922, USA). RNA was transferred from the gel onto a charged nylon membrane, UV-crosslinked to the membrane, and hybridized with P$^{32}$-labeled gene-specific oligonucleotide probes (*Supplementary file 4*) overnight at 42 °C. Blots were visualized using the Cytiva Amersham Typhoon phosphor imager. All oligonucleotide sequences used to generate probes for this study are listed in *Supplementary file 4*.

## RNA structure predictions

Secondary structures of mRNA targets were predicted in MFold (https://bio.tools/mfold) using ~60 nt sequences that encompassed the Rnt1 cleavage sites detected by PARE.

## Sequence alignments

Predicted AGNN tetraloop sequences of mRNA targets (plus 3 nts up- and downstream of the tetraloop) were aligned using WebLogo (https://weblogo.berkeley.edu/logo.cgi).

## Gene ontology analysis

Gene ontology analysis was performed for all mRNA targets using YeastEnrichr (https://maayan lab. cloud/Yeast Enrichr), PANTHER (https://pantherdb.org/), and *Saccharomyces* Genome Database Gene Ontology Term Finder (https://www.yeastgenome.org/goTermFinder).

## Confocal fluorescence microscopy

Yeast strains were grown in 10 mL cultures to late log phase (OD$_{600}$ of ~1.0) at 30 °C, then harvested by centrifugation and fixed overnight in 1% paraformaldehyde (paraformaldehyde, 1 M NaOH, 10 X PBS,

HCl to pH 7.4) at 4 °C. Cells were then harvested again, permeabilized with 0.1 X Triton for 30 min at 27 °C, and stained with NucBlue Fixed Cell Stain ReadyProbes reagent (Invitrogen, ref R37606, USA) for 1 hr at 27 °C. (Cells were washed three times in 1 X PBS prior to fixation, permeabilization, and staining.) Cells were washed and resuspended in 1 X PBS, mounted to slides using ProLong Diamond Antifade Mountant (Invitrogen, ref P36965, USA), and imaged on the Olympus Fluoview FV3000 laser scanning confocal microscope. Images were analyzed in the cellSens Dimension software.

## Experimental evolution

Thirteen 10 mL cultures of the *rnt1Δ::NEO* parent strain were grown to saturation in YPD at 30 °C, then diluted 1:1000. This was repeated 10 times, then cultures allowed to grow to mid-log phase (OD$_{600}$ of ~0.6). Strains were streaked on YPD +Neomycin plates, and cells were washed, resuspended to an OD$_{600}$ of 0.6, serially diluted (1:5), and spotted on YPD. Evolved strains showing a suppression of the *rnt1Δ* growth defect were analyzed by whole-genome sequencing.

## Protein structure modeling

Suppressor mutations in nuclear RNA exosome cofactors were modeled using the published structure of the exosome (*Schuller et al., 2018*) (RCSB PDB: 6FSZ) in the Chimera software. The Ydr514c protein structure was predicted using the AlphaFold 3 online server (*Abramson et al., 2024*) (https://alphafoldserver.com/).

## Material availability

All plasmids and yeast strains created as part of this study are available upon request from AvH.

## Acknowledgements

This work was funded by the National Institutes of Health (NIH) grant R35GM141710 to AvH and NIH grant F31GM149143 to LNS. Special thanks to Dr. Ayan Chatterjee (lab of Dr. Danielle Garsin, UT Health Science Center) for his generous assistance with fluorescence microscopy, and to members of the van Hoof and Abou Elela labs for their thoughtful comments and critiques on this manuscript.

## Additional information

### Funding

| Funder | Grant reference number | Author |
| --- | --- | --- |
| National Institute of General Medical Sciences | R35GM141710 | Lee-Ann Notice-Sarpaning Catherine Stuart |
| National Institutes of Health | F31GM149143 | Lee-Ann Notice-Sarpaning |

The funders had no role in study design, data collection and interpretation, or the decision to submit the work for publication.

### Author contributions

Lee-Ann Notice-Sarpaning, Conceptualization, Formal analysis, Validation, Investigation, Writing – original draft, Writing – review and editing; Mathieu Catala, Catherine Stuart, Investigation, Writing – review and editing; Sherif Abou Elela, Supervision, Writing – review and editing; Ambro van Hoof, Conceptualization, Data curation, Formal analysis, Supervision, Funding acquisition, Investigation, Writing – original draft, Project administration, Writing – review and editing

### Author ORCIDs

Sherif Abou Elela ⬚ https://orcid.org/0000-0002-0630-3294
Ambro van Hoof ⬚ https://orcid.org/0000-0002-7800-9764

Reviewer #1 (Public review): https://doi.org/10.7554/eLife.106662.3.sa1

Reviewer #2 (Public review): https://doi.org/10.7554/eLife.106662.3.sa2
Author response https://doi.org/10.7554/eLife.106662.3.sa3

## Additional files

### Supplementary files
Supplementary file 1. List of Rnt1 site identified.

Supplementary file 2. List of plasmids used.

Supplementary file 3. List of yeast strains used.

Supplementary file 4. List of oligos used.

MDAR checklist

### Data availability
RNA sequence data were deposited in SRA (accession number PRJNA1330880).

The following dataset was generated:

| Author(s) | Year | Dataset title | Dataset URL | Database and Identifier |
|---|---|---|---|---|
| Notice-Sarpaning L-N, Catala M, Stuart C, Abou Elela S, van Hoof A | 2025 | Mapping of in vivo cleavage sites uncovers a major role for yeast RNase III in regulating protein-coding genes | https://www.ncbi.nlm.nih.gov/bioproject/PRJNA1330880 | NCBI BioProject, PRJNA1330880 |

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
