## [Editor Report · eLife Assessment]

This **valuable** study expands the inventory of polyadenylated RNAs cleaved by the double-stranded RNA endonuclease Rnt1 in budding yeast, using **solid** methodology based on high-throughput sequencing. Previous studies had anecdotally discovered mRNA substrates, and this global characterization is comprehensive with multiple complementary controls. This study sets the stage for deeper investigations into the biological function of Rnt1 and substrate cleavage.

---

## [Referee Report · Reviewer #1 (Public review)]

Sarpaning et al. provide a thorough characterization of putative Rnt1 cleavage of mRNA in *S. cerevisiae*. Previous studies have discovered Rnt1 mRNA substrates anecdotally, and this global characterization expands the known collection of putative Rnt1 cleavage sites. The study is comprehensive, with several types of controls to show that Rnt1 is required for several of these cleavages.

Comments on revisions:

The authors have responded appropriately to the review.

---

## [Referee Report · Reviewer #2 (Public review)]

This study presents a useful inventory of polyadenylated RNAs cleaved by the double-stranded RNA endonuclease Rnt1 in yeast. The data were obtained with solid methodology based on high-throughput sequencing, and the evidence that Rnt1 contributes to cellular homeostasis through controlling the turnover of selected mRNAs is convincing.

Comments on revisions:

I appreciate the authors' thorough and thoughtful response, and I find that the manuscript has been substantially strengthened by the additional data, analyses, and textual clarifications.

---

## [Author Response]

The following is the authors’ response to the original reviews.

**Reviewer #1 (Public review):**
Strengths:Sarpaning et al. provide a thorough characterization of putative Rnt1 cleavage of mRNA in *S. cerevisiae*. Previous studies have discovered Rnt1 mRNA substrates anecdotally, and this global characterization expands the known collection of putative Rnt1 cleavage sites. The study is comprehensive, with several types of controls to show that Rnt1 is required for several of these cleavages.Weaknesses:(1) Formally speaking, the authors do not show a direct role of Rnt1 in mRNA cleavage - no studies were done (e.g., CLIP-seq or similar) to define direct binding sites. Is the mutant Rnt1 expected to trap substrates? Without direct binding studies, the authors rely on genetics and structure predictions for their argument, and it remains possible that a subset of these sites is an indirect consequence of rnt1. This aspect should be addressed in the discussion.

We have added to this point in the discussion, as requested. We do not, however, agree that CLIP-seq or other methods are needed to address this point, or would even be helpful in the question the reviewer raises.

Importantly, we show that recombinant Rnt1 purified from *E. coli* cleaves the same sites as those mapped in vivo. This does provide direct evidence that Rnt1 directly binds those RNAs. Furthermore, it shows that it can bind these RNAs without the need of other proteins. Our observation that many mRNAs are cleaved at -14 and +16 positions from NGNN stem loops to leave 2-nt 3’ overhangs provides further support that these are the products of an RNase III enzyme, and Rnt1 is the only family member in yeast. Thus, we disagree with the reviewer that our studies do not show direct targeting.

CLIP-seq experiments would be valuable, but they would address a different point. CLIP-seq measures protein binding to RNA targets, and it is likely that Rnt1 binds some RNAs without cleaving them. In addition, only a transient interaction are needed for cleavage and such transient interactions might not be readily detected by CLIP-seq. Thus, CLIP-seq would reveal the RNAs bound by Rnt1, but would not help identify which ones are cleaved. Catala et al (2004) showed that the catalytically inactive mutant of Rnt1 carries out some functions that are important for the cell cycle. The CLIP-seq studies would be valuable to determine these non-catalytic roles of Rnt1, but we consider those questions beyond the scope of the current study.

(2) The comprehensive list of putative Rnt1 mRNA cleavage sites is interesting insofar as it expands the repertoire of Rnt1 on mRNAs, but the functional relevance of the majority of these sites remains unknown. Along these lines, the authors should present a more thorough characterization of putative Rnt1 sites recovered from in vitro Rnt1 cleavage.

We have included new data that confirm that YDR514C cleavage by Rnt1 is relevant to yeast cell physiology. We show that YDR514C overexpression is indeed toxic, as we previously postulated. More importantly, we generated an allele of YDR514C that has synonymous mutations designed to disrupt the stem-loop recognized by Rnt1. We show that at 37 °C, both the wild-type and mutant allele are toxic to rnt1∆ cells, but that in cells that express Rnt1, the wild-type cleavable allele is more toxic than the allele with the mutated stem-loop. This genetic interaction provides strong evidence that cleavage of YDR514C by Rnt1 is relevant to cell physiology.

We have also added PARE analysis of poly(A)-enriched and poly(A)-depleted reactions and show that compared to Dcp2, Rnt1 preferentially targets poly(A)+ mRNAs, consistent with it targeting nuclear RNAs. We discuss in more detail that by cleaving nuclear RNA, Rnt1 provides a kinetic proofreading mechanism for mRNA export competence.

(3) The authors need to corroborate the rRNA 3'-ETS tetraloop mutations with a northern analysis of 3'-ETS processing to confirm an ETS processing defect (which might need to be done in decay mutants to stabilize the liberated ETS fragment). They state that the tetraloop mutation does not yield a growth defect and use this as the basis for concluding that rRNA cleavage is not the major role of Rnt1 in vivo, which is a surprising finding. But it remains possible that tetraloop mutations did not have the expected disruptive effect in vivo; if the ETS is processed normally in the presence of tetraloop mutations, it would undermine this interpretation. This needs to be more carefully examined.

We have removed the rRNA 3'-ETS tetraloop mutations, because initial northern blot analysis indicated that Rnt1 cleavage is not completely blocked by the mutations we designed. Therefore, the reviewer is correct that tetraloop mutations did not have the expected disruptive effect in vivo. Future investigations will be required to fully understand this. This was a minor point and removing this focuses the paper on its major contributions

(4) To support the assertion that YDR514C cleavage is required for normal "homeostasis," and more specifically that it is the major contributor to the rnt1∆ growth defect, the authors should express the YDR514C-G220S mutant in the rDNA∆ strains with mutations in the 3'-ETS (assuming they disrupt ETS processing, see above). This simple experiment should provide a relative sense of "importance" for one or the other cleavage being responsible for the rnt1∆ defect. Given the accepted role of Rnt1 cleavage in rRNA processing and a dogmatic view that this is the reason for the rnt1∆ growth defect, such a result would be surprising and elevate the functional relevance and significance of Rnt1 mRNA cleavage.

We agree that the experiment proposed by the reviewer is very simple, but we are puzzled by the rationale. First, our experiments do not support that there is anything special about the G220S mutation in YDR514C. A complete loss of function (ydr514c∆) also suppresses the growth defect, suggesting that ydr514c-G220S is a simple loss of function allele. We have clarified that the G220S mutation is distant from the stem-loop recognized by Rnt1 and is unlikely to affect cleavage by Rnt1. Instead, Rnt1 cleavage and the G220S mutation are independent alternative ways to reduce Ydr514c function. We have clarified this point in the text.

As mentioned in response to point #3, we have included other additional experiments that address the same overall question raised here – the importance of YDR514C mRNA cleavage by Rnt1.

(5) Given that some Rnt1 mRNA cleavage is likely nuclear, it is possible that some of these targets are nascent mRNA transcripts, as opposed to mature but unexported mRNA transcripts, as proposed in the manuscript. A role for Rnt1 in co-transcriptional mRNA cleavage would be conceptually similar to Rnt1 cleavage of the rRNA 3'-ETS to enable RNA Pol I "torpedo" termination by Rat1, described by Proudfoot et al (PMID 20972219). To further delineate this point, the authors could e.g., examine the poly-A tails on abundant Rnt1 targets to establish whether they are mature, polyadenylated mRNAs (e.g., northern analysis of oligo-dT purified material). A more direct test would be PARE analysis of oligo-dT enriched or depleted material to determine the poly-A status of the cleavage products. Alternatively, their association with chromatin could be examined.

We have added the requested PARE analysis of oligo-dT enriched or depleted material to determine the polyA status of the cleavage products and related discussions. These confirm our proposal that Rnt1 cleaves mature but unexported mRNA transcripts

We also note that the northern blots shown in figures 2E, 4C, and 5B use oligo dT selected RNA because the signal was undetectable when we used total RNA. This suggests that the cleaved mRNAs are indeed polyadenylated.

The term nascent is somewhat ambiguous, but if the reviewer means RNA that is still associated with Pol II and has not yet been cleaved by the cleavage and polyadenylation machinery, we think that is inconsistent with our findings. We have also re-analyzed the NET-seq data from https://pubmed.ncbi.nlm.nih.gov/21248844/ and find no prominent peaks for our Rnt1 sites in Pol II associated RNAs, although for BDF2 NET-seq does suggest that “spliceosome-mediated decay” is co-transcriptional as would be expected. Altogether these data confirm our previous proposal that Rnt1 mainly cleaves mRNAs that have completed polyadenylated but are not yet exported.

(6) While laboratory strains of budding yeast have a single RNase III ortholog Rnt1, several other budding yeast have a functional RNAi system with Dcr and Ago (PMID 19745116), and laboratory yeast strains are a derived state due to pressure from the killer virus to lose the RNAi system (PMID 21921191). The current study could provide new insight into the relative substrate preferences of Rnt1 and budding yeast Dicer, which could be experimentally confirmed by expressing Dcr in RNT1 and rnt1∆ strains. In lieu of experiments, discussion of the relevance of Rnt1 cleavage compared to yeast RNAi should be included in the discussion before the "human implications" section.

The reviewer points out that most other eukaryotic species have multiple RNase III family members, which is a general point we discussed and have now expanded on. The reviewer specifically points to papers that study a species that was incorrectly referred to as Saccharomyces castellii in PMID 19745116, but whose current name is Naumovozyma castellii, reflecting that it is not that closely related to *S. cerevisiae* (diverged about 86 million years ago; for the correct species phylogeny, see http://ygob.ucd.ie/browser/species.html, as both of the published papers the reviewer cites have some errors in the phylogeny).

The other species discussed in PMID 19745116 (Vanderwaltozyma polyspora and Candida albicans) are even more distant. There have been several studies on substrate specificity of Dcr1 versus Rnt1 (including PMID 19745116).

The reviewer suggests that expressing Dcr1 in *S. cerevisiae* would be a valuable addition. However, we can’t envision a mechanism by which *S. cerevisiae* maintained physiologically relevant Dcr1 substrates in the absence of Dcr1. The results from the proposed study would, in our opinion, be limited to identifying RNAs that can be cleaved in this particular artificial system. We think an important implication of our work is that similar studies to ours should be caried out in rnt1∆, dcr1∆, and double mutants in either *S. pombe* or *N. castellii*, as well as in drosha knock outs in animals, and we discuss this in more detail in the revised paper.

(7) For SNR84 in Figure S3D, it appears that the TSS may be upstream of the annotated gene model. Does RNA-seq coverage (from external datasets) extend upstream to these additional mapped cleavages? The assertion that the mRNA is uncapped is concerning; an alternative explanation is that the nascent mRNA has a cap initially but is subsequently cleaved by Rnt1. This point should be clarified or reworded for accuracy.

We agree with the reviewer that the most likely explanation is that the primary SNR84 transcript is capped, and 5’ end processed by Rnt1 and Rat1 to make a mature 5’ monophosphorylated SNR84 and have clarified the text accordingly. We suspect our usage of “uncapped” might have been confusing. “uncapped” was not meant to indicate that the primary transcript did not receive a cap, but instead that the mature transcript did not have a cap. We now use “5’ end processed” and “5’ monophosphorylated”.

**Reviewer #2 (Public review):**
The yeast double-stranded RNA endonuclease Rnt1, a homolog of bacterial RNase III, mediates the processing of pre-rRNA, pre-snRNA, and pre-snoRNA molecules. Cells lacking Rnt1 exhibit pronounced growth defects, particularly at lower temperatures. In this manuscript, Notice-Sarpaning examines whether these growth defects can be attributed at least in part to a function of Rnt1 in mRNA degradation. To test this, the authors apply parallel analysis of RNA ends (PARE), which they developed in previous work, to identify polyA+ fragments with 5' monophosphates in RNT1 yeast that are absent in rnt1Δ cells. Because such RNAs are substrates for 5' to 3' exonucleolytic decay by Rat1 in the nucleus or Xrn1 in the cytoplasm, these analyses were performed in a rat1-ts xrn1Δ background. The data recapitulate known Rtn1 cleavage sites in rRNA, snRNAs, and snoRNAs, and identify 122 putative novel substrates, approximately half of which are mRNAs. Of these, two-thirds are predicted to contain double-stranded stem loop structures with A/UGNN tetraloops, which serve as a major determinant of Rnt1 substrate recognition. Rtn1 resides in the nucleus, and it likely cleaves mRNAs there, but cleavage products seem to be degraded after export to the cytoplasm, as analysis of published PARE data shows that some of them accumulate in xrn1Δ cells. The authors then leverage the slow growth of rnt1Δ cells for experimental evolution. Sequencing analysis of thirteen faster-growing strains identifies mutations predominantly mapping to genes encoding nuclear exosome co-factors. Some of the strains have mutations in genes encoding a laratdebranching enzyme, a ribosomal protein nuclear import factor, poly(A) polymerase 1, and the RNAbinding protein Puf4. In one of the puf4 mutant strains, a second mutation is also present in YDR514C, which the authors identify as an mRNA substrate cleaved by Rnt1. Deletion of either puf4 or ydr514C marginally improves the growth of rnt1Δ cells, which the authors interpret as evidence that mRNA cleavage by Rnt1 plays a role in maintaining cellular homeostasis by controlling mRNA turnover.While the PARE data and their subsequent in vitro validation convincingly demonstrate Rnt1mediated cleavage of a small subset of yeast mRNAs, the data supporting the biological significance of these cleavage events is substantially less compelling. This makes it difficult to establish whether Rnt1-mediated mRNA cleavage is biologically meaningful or simply "collateral damage" due to a coincidental presence of its target motif in these transcripts.

We thank the reviewer and have added additional data to support our conclusion that mRNA cleavage, at least for YDR514C, is not simply collateral damage, but a physiologically relevant function of Rnt1. From an evolutionary perspective, cleavage of mRNAs by Rnt1 might have initially been collateral damage, but if there is a way to use this mechanism, evolution is probably going to use it.

(1) A major argument in support of the claim that "several mRNAs rely heavily on Rnt1 for turnover" comes from comparing number of PARE reads at the transcript start site (as a proxy for fraction of decapped transcripts) and at the Rnt1 cleavage site (as a proxy for fraction of Rnt1-cleaved transcripts). The argument for this is that "the major mRNA degradation pathway is through decapping". However, polyA tail shortening usually precedes decapping, and transcripts with short polyA tails would be strongly underrepresented in PARE sequencing libraries, which were constructed after two rounds of polyA+ RNA selection. This will likely underestimate the fraction of decapped transcripts for each mRNA. There is a wide range of well-established methods that can be used to directly measure differences in the half-life of Rnt1 mRNA targets in RNT1 vs rnt1Δ cells. Because the PARE data rely on the presence of a 5' phosphate to generate sequencing reads, they also cannot be used to estimate what fraction of a given mRNA transcript is actually cleaved by Rnt1.

The reviewer is correct that decapping preferentially affects mRNAs with shortened poly(A) tails, that Rnt1 cleavage likely affects mostly newly made mRNAs with long poly(A) tails, and that PARE may underestimate the decay of mRNAs with shortened poly(A) tails. We have reanalyzed our previously published data where we performed PARE on both the poly(A)-enriched fraction and the poly(A)-depleted fraction (that remains after two rounds of oligo dT selection). Rnt1 products are over-represented in the poly(A)-enriched fraction, while decapping products are enriched in the poly(A)-depleted fraction, providing further support to our conclusion that Rnt1 cleaves nuclear RNA. We have re-written key sections of the paper accordingly.

The reviewer also points out that “There is a wide range of well-established methods that can be used to directly measure differences in the half-life of Rnt1 mRNA targets in RNT1 vs rnt1Δ cells.” However, all of those methods measure mRNA degradation rates from the steady state pool, which is mostly cytoplasmic. We have, in different contexts, used these methods, but as we pointed out they are inappropriate to measure degradation of nuclear RNA. There are some studies that measure nuclear degradation rates, but this requires purifying nuclei. There are two major drawbacks to this. First, it cannot distinguish between degradation in the nucleus and export from the nucleus because both processes cause disappearance from the nucleus. Second, the purification of yeast nuclei requires “spheroplasting” or enzymatically removing the rigid cell wall. This spheroplasting is likely to severely alter the physiological state of the yeast cell. Given these significant drawbacks and the substantial time and money required, we chose not to perform this experiment.

(2) Rnt1 is almost exclusively nuclear, and the authors make a compelling case that its concentration in the cytoplasm would likely be too low to result in mRNA cleavage. The model for Rnt1-mediated mRNA turnover would therefore require mRNAs to be cleaved prior to their nuclear export in a manner that would be difficult to control. Alternatively, the Rnt1 targets would need to re-enter prior to cleavage, followed by export of the cleaved fragments for cytoplasmic decay. These processes would need to be able to compete with canonical 5' to 3' and 3' to 5' exonucleolytic decay to influence mRNA fate in a biologically meaningful way.

We disagree that mRNA export would be difficult to control, as is elegantly demonstrated by the 13 KDa HIV Rev protein. The export of many other RNAs is tightly controlled such that many RNAs are rapidly degraded in the nucleus by, for example, Rat1 and the RNA exosome, while other RNAs are rapidly exported. Indeed, the competition between RNA export and nuclear degradation is generally thought to be an important quality control for a variety of mRNAs and ncRNAs. We do agree with the reviewer that re-import of mRNAs appears unlikely (which is why we do not discuss it), although it occurs efficiently for other Rnt1-cleaved RNAs such as snRNAs. We have clarified the text accordingly, including in the introduction, results, and discussion.

(3) The experimental evolution clearly demonstrates that mutations in nuclear exosome factors are the most frequent suppressors of the growth defects caused by Rnt1 loss. This can be rationalized by stabilization of nuclear exosome substrates such as misprocessed snRNAs or snoRNAs, which are the major targets of Rnt1. The rescue mutations in other pathways linked to ribosomal proteins (splicing, ribosomal protein import, ribosomal mRNA binding) support this interpretation. By contrast, the potential suppressor mutation in YDR514C does not occur on its own but only in combination with a puf4 mutation; it is also unclear whether it is located within the Rnt1 cleavage motif or if it impacts Rnt1 cleavage at all. This can easily be tested by engineering the mutation into the endogenous YDR514C locus with CRISPR/Cas9 or expressing wild-type and mutant YDR514C from a plasmid, along with assaying for Rnt1 cleavage by northern blot. Notably, the growth defect complementation of YDR514C deletion in rnt1Δ cells is substantially less pronounced than the growth advantage afforded by nuclear exosome mutations (Figure S9, evolved strains 1 to 5). These data rather argue for a primary role of Rnt1 in promoting cell growth by ensuring efficient ribosome biogenesis through pre-snRNA/pre-snoRNA processing.

The reviewer makes several points.

First, we have clarified that the ydr514c-G220S mutation is not near the Rnt1 cleavage motif and is unlikely to affect cleavage by Rnt1. This is exactly what would be expected for a mutation that was selected for in an rnt1∆ strain. Although the reviewer appears to expect it, a mutation that affects Rnt1 cleavage could not be selected for in a strain that lacks Rnt1.

Second, the reviewer points out that the original ydr514c mutations arose in a strain that also had a puf4 deletion. However, we show that ydr514c∆ also suppresses rnt1∆. Furthermore, we have added additional data that overexpressing an uncleavable YDR514C mRNA affects yeast growth at 37 °C more than the wild-type cleavable form further supporting that the cleavage of YDR154C by Rnt1 is physiologically relevant.

**Reviewer #2 (Recommendations for the authors):**
(1) The description of the PARE library construction protocol and data analysis workflow is insufficient to ensure their robustness and reproducibility. The library construction protocol should include details of the individual steps, and the data analysis workflow description should include package versions and exact commands used for each analysis step.

We have clarified that the experiments were performed exactly as previously described and have included very detailed methods. The Galaxy server does not require commands and instead we have indicated the parameters chosen in the various steps. We have also added that the PARE libraries for poly(A)+ and poly(A)- fractions were generated in the lab of Pam Green according to their protocol, which is not exactly the same as ours. Nevertheless, the Rnt1 sites are also evident from those libraries, further demonstrating the robustness of our data.

(2) PARE signal is expressed as a ratio of sequencing coverage at a given nucleotide in RNT1 vs rnt1Δ cells. This poses challenges to estimating fold changes: by definition, there should be no coverage at Rnt1 cleavage sites in rnt1Δ cells, as there will not be any 5' monophosphate-containing mRNA fragments to be ligated to the library construction linker. This should be accounted for in the data analysis pipeline - the DESeq2 package, for example, handles this very well (https://support.bioconductor.org/p/64014/).

The reviewer is correct and we have clarified how we do account for the possibility of having 0 reads by adding an arbitrary 0.01 cpm to all PARE scores for wild type and mutant. In the original manuscript this was not explicitly mentioned and the reader would have to go to our previous paper to learn about this detail. Adding this 0.01 cpm pseudocount avoids dividing by 0 when we calculate a comPARE score. This means we actually underestimate the fold change. As can be seen in the red line in the image below, the y-axis modified log2FC score maxes out along a diagonal line at log2([average RNT1 reads]/0.01) instead of at infinity. That is, at a wild type peak height of 1 cpm, the maximum possible score is log2(1.01/.01), which equals 6.66, and at 10 cpm, the maximum score is ~10, etc.. As can be seen, many of the scores fall along this diagonal, reflecting that indeed, there are 0 reads in the rnt1∆ samples.

There are multiple ways to deal with this issue, and ours is not uncommon. DESeq2, suggested by the reviewer, uses a different method, which relies on the assumption that the dispersion of read counts for genes of any given expression strength is constant, and then uses that dispersion to “correct” the 0 read counts. While this is a valid way for differential gene expression when comparing similar RNAs, the underlying assumption that the dispersion of expression of all genes is similar for similar expression level is questionable for comparing, for example, mRNAs, snoRNAs, and snRNAs. Thus, we are not convinced that this is a better way to deal with 0 counts. Our analysis accepts that 0 might be the best estimate for the number of counts that are expected from rnt1∆ samples.

(3) The analysis in Figure S8 is insufficient to demonstrate that the four mRNAs depicted are significantly more abundant in rnt1Δ vs RNT1 cells - differences in coverage could simply be a result of different sequencing depth. Please use an appropriate method for estimating differential expression from RNA-Seq data (e.g., DESeq2).

Unfortunately, the previously published data we included as figure S8 (now figure S9) did not include replicates, and we agree that it does not rigorously show an effect. The reviewer suggests that we analyze the data by DESeq2, which requires replicates, and thus, cannot be done. Instead we have clarified this. If the reviewer is not satisfied with this, we are prepared to delete it.